

# δ13C values in stalagmites from tropical South America for the last two millennia

Valdir Felipe Novello[1]*; Francisco William da Cruz[1], Mathias Vuille[2], José Leandro Pereira Silveira Campos[1], Nicolás Misailidis Stríkis[3], James Apaéstegui[4], Jean Sebastien Moquet[5], Vitor Azevedo[1], Angela Ampuero[3], Giselle Utida[1], Xianfeng Wang[6], Gustavo Macedo Paula-Santos[7], Plinio Jaqueto[8], Luiz Carlos Ruiz Pessenda[9], Daniel O Breecker[10], Ivo Karmann[1]

[1]Institute of Geosciences, University of São Paulo, São Paulo, 05508-080, Brazil.
[2]Department of Atmospheric and Environmental Sciences, University at Albany, Albany, 2222, USA.
[3]Departamento de Geoquímica, Universidade Federal Fluminense, Niterói, 24020-141, Brazil.
[4]Instituto Geofísico del Perú, Lima, 15012, Peru.
[5]Institut des Sciences de la Terre d'Orléans, Orléans, 45100, France.
[6]Earth Observatory of Singapore, Nanyang Technological University, Jurong West, 639798, Singapore.
[7]Institute of geoscience Umiversity of Campinas, 13083-855.
[8]Instituto de Astronomia, Geofísica e Ciências Atmosféricas, University of São Paulo, São Paulo, 05508-090, Brazil
[9]Center for Nuclear Energy in Agriculture (CENA), University of São Paulo, São Paulo, 13416-000, Brazil.
[10]Jackson School of Geosciences, University of Texas, Austin, 2305, USA.

*Correspondence to: Valdir F. Novello (vfnovello@gmail.com).

## Abstract

Due to the many factors controlling δ13C values in stalagmites, complicating their paleoclimatic and paleoenvironmental interpretation, most studies do not present δ13C values, but instead focus mainly on δ18O values. This is also the case for most cave studies from tropical South America, where many new δ18O stalagmite records covering the last millennia were recently published. Here, we test the influence of local hydroclimate, altitude, temperature and changing vegetation types on δ13C values in stalagmites, by employing a new dataset (named δ13C_2k_SA - Novello et al., 2020, https://doi.pangaea.de/10.1594/PANGAEA.919050) composed of published and unpublished carbon isotope records from various sites in tropical South America. Most locations were dominated by C3 plants over the last two millennia and are characterized by speleothem δ13C values more depleted than -6 ‰. The main factors influencing δ13C values are associated with the local hydroclimate, followed by minor effects from temperature. Most of the isotopic records show a significant correlation between the δ13C and δ18O values, indicating a close relationship between local hydroclimate and atmospheric convective processes related to the South American Monsoon System.



## 1 Introduction

Measurements of carbon isotope ($\delta^{13}$C) ratios are essential for (paleo)environmental studies, such as those regarding the carbon cycle, past food consumption by pre-historic societies, paleo-vegetation

reconstructions, soil dynamics, aspects regarding animal migration and food consumption, etc. (Rayner et al., 1999; Calo and Cortés, 2009; Pansani et al., 2019; Novello et al., 2019). However, $\delta^{13}$C values of speleothems are generally considered difficult to interpret and, in most cases, are not reported in isotopic studies based on speleothems. Studies from tropical South America, in particular, have almost completely neglected $\delta^{13}$C values from stalagmites and instead relied solely on $\delta^{18}$O values for reconstruction of the

South American Monsoon System (SAMS) (Reuter et al., 2009; Novello et al., 2012; 2016; 2018; Apaéstegui et al., 2014; 2018; Vuille et al., 2012; Kanner et al., 2013; Wang et al., 2017). Here we present a compilation of 25 $\delta^{13}$C records (13 of them hitherto unpublished) from stalagmites collected at different sites throughout tropical South America and covering the last two millennia (referred to here as the $\delta^{13}$C_2k_SA dataset) with aims to characterize the main factors controlling $\delta^{13}$C variability in these

stalagmites and provide possible paleoclimate and paleoenvironmental reconstructions for the region.
In Section 2 we review the mechanisms that control the $\delta^{13}$C values in the stalagmites and the related global and regional forcings. In Section 3 we present the data, the stalagmites and methods. In Section 4 we present the study sites and describe the environmental and climate characteristics that might affect the $\delta^{13}$C values in the stalagmites from tropical South America. In Section 5 we show the results (data set) and in Section

6 we discuss the interpretations of the new records and their regional relationships. Additionally, using Principal Component Analysis (PCA), we propose a new hydrological and vegetation index for South America based on the $\delta^{13}$C_2k_SA dataset.

## 2. Background

### 2.1 Main processes controlling $\delta^{13}$C values in speleothems

The initial source of carbon for speleothems is the soil $CO_2$ and tree roots. It forms carbonic acid in contact with water, which dissolves cave host-rock (limestones or dolostones) according to the equation:

$$Eq. \quad CaCO_3 + H_2O + CO_2 \leftrightarrow Ca^{2+} + 2HCO_3^- \quad (1)$$



Open and closed system models were proposed to explain the dissolution of calcium carbonate in the percolating solution (Fohlmeister et al., 2011; McDermott et al., 2004, Hendy, 1971). Initially, the percolating solution remains in equilibrium with the infinite reservoir of soil $CO_2$ and, thereby, the bicarbonate in solution receives its $\delta^{13}C$ fingerprint. Under these conditions, the contribution of the $\delta^{13}C$ values from the bedrock to the $HCO_3^-$ in solution can be neglected. In a closed system, during the percolation into the epikarst, the solution loses contact with the soil $CO_2$, and the $CO_2$ in solution is progressively consumed through the dissolution of the bedrock. The rock dissolution is limited by the initial amount of $CO_2$ and, consequently, through this process the $\delta^{13}C$ from the bedrock influences the isotopic composition of the remaining solution (McDermott, 2004). In most caves, the interaction between the percolation solution and the host-rock occurs as a partially open system. Thus, carbon in speleothems is sourced mainly from soil organic matter and tree roots, with a small proportion coming from the bedrock (usually less than 10%) (Genty et al., 2001).

The $\delta^{13}C$ values in pedogenic carbonate are closely related to the isotopic values from the surrounding vegetation (Cerling, 1984; Quade et al., 1989), defined by the plant type ($C_3$, $C_4$ or CAM), which in turn, will be conditioned by climatic parameters such as temperature, pluviosity, rainfall seasonality and atmospheric $CO_2$ (Ehleringer et al., 1997). Vegetation dominated by $C_3$ plants has $\delta^{13}C$ values between -32 ‰ and -20 ‰, while vegetation dominated by $C_4$ plants is characterized by values between -17 ‰ to -9 ‰ (Badeck et al., 2005). CAM plants have $\delta^{13}C$ values that overlap with both $C_3$ and $C_4$ plants. In addition, $\delta^{13}C$ values of individual $C_3$ plant species can vary approximately 1 to 2 ‰, depending on water availability (e.g., Hartman and Danin, 2010). The differences in the $\delta^{13}C$ values from total organic matter in soil resulting from the dominant plant types are transferred to stalagmites precipitated under their respective environment. Thus, stalagmites precipitated in caves under conditions dominated by $C_3$ plants typically have values ranging from -14 ‰ to -6 ‰, while those forming below $C_4$ plant cover range from -6 % to +2 % (McDermott, 2004; Baker et al., 1997; Dreybrodt, 1988). Variations in soil $\delta^{13}C$ values and their evolution over time are controlled by carbon inputs from vegetation, which is proportional to the organic matter amount and vegetation density; thus denser vegetation is also associated with more depleted values in soil $\delta^{13}C$ and vice-versa (Pessenda et al., 2010). In addition, the $\delta^{13}C$ values of $C_3$ plants are sensitive to atmospheric $CO_2$ levels (Van de Water et al., 1994; Schibert and Jahen 2012), producing a signal that is transferred to stalagmites (Breecker, 2017).



In the cave system, $\delta^{13}C$ values from dissolved inorganic carbon (DIC) can undergo fractionation through prior calcite precipitation (PCP), which preferentially removes $^{12}C$ from the solution during precipitation forced by $CO_2$ degassing (Micker et al., 2019), depleting $^{12}C$ from the final isotopic product recorded in stalagmites (Baker et al., 1997). PCP increases during drier periods due to the increased exposure of seepage solution to air pockets along the epikarst flow routes, which results in $CO_2$ degassing from the solution, promoting the carbonate precipitation in the epikarst and/or stalactites (Fairchild and Baker, 2012). Therefore, PCP is climate related. In monsoonal climates, such as in (sub)tropical South America, the amount effect is the dominant process that affects the $\delta^{18}O$ value of rainwater, such that an increase in the amount of rainfall results in waters with lower $\delta^{18}O$ values (Vuille et al., 2012). This increase in rainfall amount might also cause a decrease in the $\delta^{13}C$ value of speleothem calcite, by increasing the soil moisture content and soil respiration rates or by reducing PCP, resulting in a positive correlation between the $\delta^{18}O$ and $\delta^{13}C$ values of speleothem calcite/aragonite (Cruz et al., 2006; Mickler et al., 2006; and references therein). While $\delta^{13}C$ and $\delta^{18}O$ values may be correlated for different reasons, high correlation between both isotopic ratios can also be indicative of forced kinetic fractionation, since carbon and oxygen are fractioned in the same direction in this process (Hendy et al., 1971). The isotopic disequilibrium increases with enhanced ventilation of the cave, which depends on temperature. However, the range of values resulting from the fractionation factor between $CaCO_3$ and $HCO_3^-$ in tropical temperatures of 15 – 30 °C has been documented to be < 0.5 ‰ (Polag et al., 2010 and references therein). Recently, Fohlmeister et al. (2020) using a large dataset of $\delta^{13}C$ values in stalagmites deposited post-1900 CE show evidence for a temperature control on this proxy, likely driven by vegetation and soil processes, while PCP can explain the wide $\delta^{13}C$ range observed for concurrently deposited samples from the same cave.

### 2.2 Processes controlling the $\delta^{13}C$ values in speleothems from South America

The main mode of climate variability in tropical South America is defined by the SAMS behavior, which also influences vegetation changes. Most of the $\delta^{18}O$ records from stalagmites in South America show a weak monsoon activity during the Medieval Climate Anomaly (MCA, 900-1100 years CE), in contrast with a strong activity during the Little Ice Age (LIA, 1600-1850 years CE) (Bird et al., 2011; Vuille et al., 2012; Novello et al., 2016; 2018). During the LIA period, a moisture dipole was documented between east and west portions of tropical South America, due to the displacement of the South Atlantic Convergence Zone (SACZ) toward the southwest (Campos et al., 2019; Novello et al., 2018). The implications of this change





in climate on vegetation and local hydroclimate, which are parameters that heavily influence the $\delta^{13}C$ values in stalagmites, are still not well studied.

Few studies focused on the $\delta^{13}C$ values from speleothems in South America. Cruz et al. (2006) interpreted the $\delta^{13}C$ from Botuverá cave (southeastern Brazil) as a proxy for soil $CO_2$ productivity, which is modulated

at orbital time scales by changes in local temperature brought about by shifts between summer monsoonal and winter extratropical circulation. Novello et al. (2019) reported a decrease of 9 ‰ in the $\delta^{13}C$ values from Jaraguá cave (one of the caves of this study) during the transition between the last Glacial and the Holocene periods, resulting from a combination of changes above the cave, including: changes in the predominant vegetation type from $C_4$ to $C_3$, increase of organic matter and soil horizons, which were mainly

caused by the increase in temperature and atmospheric $CO_2$. For the last two millennia, Jaqueto et al. (2016) show that the concentration of magnetic minerals and $\delta^{13}C$ values in the ALHO6 stalagmite from Pau d'Alho cave is governed by soil dynamics and changes in vegetation cover above the cave. Dry periods are associated with less stable soils, resulting in high erosion and increased mineral flux into karst systems, which occurs simultaneously with increasing $\delta^{13}C$ values. Conversely, wetter periods with low $\delta^{13}C$ values

are associated with soils topped by denser vegetation that retains micrometer-scale pedogenic minerals and thus reduces detrital fluxes into the cave. Azevedo et al. (2019) further explored the relationship between $\delta^{13}C$ and $\delta^{18}O$ during the last millennium in a speleothem (MV3) from the Brazilian Central region, where wetter (drier) periods are also associated with lower (higher) values of $\delta^{18}O$ and $\delta^{13}C$ and higher (lower) speleothem growth rates. The same interpretation was utilized for the $\delta^{13}C$ values of the stalagmites from

Tamboril cave, located further south. However, at this site the $\delta^{18}O$ and $\delta^{13}C$ do not co-vary, which the authors interpret as a result of the decoupling between the local hydroclimate and SAMS, documented by the $\delta^{13}C$ and $\delta^{18}O$, respectively (Wortham et al., 2017).

The global increase of the atmospheric $CO_2$ concentration and temperature likely also influenced the $\delta^{13}C$ values of stalagmites during the last century. Since 1850 the land surface air temperature increased by about

1.44 °C (Jia et al., in press), affecting the fractionation between the seepage solution and the carbonate precipitated inside the caves (Lachniet, 2009), as well as promoting changes in cave ventilation (Baldini et al., 2008). Furthermore, depleted isotopic carbon has been emitted into the atmosphere due to the burning of fossil fuels, thereby decreasing the atmospheric $\delta^{13}C$ that is transferred to vegetation. The increase of atmospheric $CO_2$ can also favor the flourishing of $C_3$ plants, which are less adapted than $C_4$ plants to low

atmospheric $CO_2$ concentrations (Ehleringer et al., 1997).



## 3 Data Material and Methods

### 3.1 δ¹³C data

The dataset used in this study comprises 25 speleothem δ¹³C records, of which 13 were hitherto unpublished (Novello et al., 2020, available at https://doi.pangaea.de/10.1594/PANGAEA.919050). δ¹⁸O records and

chronological models have been published for all stalagmite records. Data from the published records was obtained from the supplementary material of the respective papers or downloaded from the NOAA speleothem database (http://www.ncdc.noaa.gov/data-access/paleoclimatology-data/datasets/speleothem). Our main goal with this study is to introduce new cave records to the speleothem community and contextualize their interpretation in a regional framework for South America during the last two millennia.

For that, we used only datasets that were available to us with speleothem δ¹³C data from the last two thousand years (since year 0 CE). Therefore, we only consider this time period for the analyses presented here. All records used, including their references, cave names, locations and climatic parameters, are listed in **Table 1**. They are composed of 11,601 δ¹³C values from 25 speleothems, published in 15 different papers (**Table 1**).


| Stalagmite | Cave | Latitude °N | Longitude °E | Reference | Annual Precipitation (mm) | Mean annual T (°C) | Elevation (m.a.s.l) |
|---|---|---|---|---|---|---|---|
| JAR4 | Jaraguá | -21.08 | -56.58 | Novello et al. (2018; 2019) This study | 1400 | 21.4 | 570 |
| JAR1 | Jaraguá | -21.08 | -56.58 | Novello et al. (2018) This study | 1400 | 21.4 | 570 |
| SBE3 | São Bernardo | -13.81 | -46.35 | Novello et al. (2018) This study | 1270 | 23.0 | 630 |
| SMT5 | São Matheus | -13.81 | -46.35 | Novello et al. (2018) This study | 1270 | 23.0 | 630 |
| ALHO6 | Pau d'Alho | -15.21 | -56.8 | Novello et al. (2016) Jaqueto et al. (2016) | 1440 | 25.5 | 340 |
| CUR4 | Curupira | -15.02 | -56.78 | Novello et al. (2016) This study | 1440 | 25.5 | 340 |
| DV2 | Diva | -12.37 | -41.57 | Novello et al. (2012) This study | 700 | 24.5 | 480 |
| TR5 | Torrinha | -12.37 | -41.57 | Novello et al. (2012) This study | 700 | 24.5 | 480 |
| LD12 | Lapa Doce | -12.37 | -41.57 | Novello et al. (2012) This study | 700 | 24.5 | 480 |
| TRA7 | Trapiá | -5.59 | -37.70 | Utida et al. (2020) | 700 | 28.0 | 72 |
| FN1 | Furna Nova | -5.60 | -37.44 | Cruz et al. (2009) Utida et al. (2020) | 700 | 28.0 | 100 |
| TM0 | Tamboril | -16.80 | -47.27 | Wortham et al. (2017) | 1400 | 22.5 | 600 |
| ANJOS1 | Lapa dos Anjos | -14.39 | -44.30 | Stríkis (2015) This study | 940 | 23.7 | 640 |
| BTV21a | Botuverá | -27.21 | -49.09 | Bernal et al. (2016) This study | 1400 | 22.0 | 200 |
| PAR3 | Paraiso | -4.07 | -55.45 | Wang et al. (2017) | 2400 | 26.0 | 60 |
| PAR1 | Paraiso | -4.07 | -55.45 | Wang et al. (2017) | 2400 | 26.0 | 60 |
| MV3 | Mata Virgem | -11.62 | -47.49 | Azevedo et al. (2019) | 1570 | 26.8 | 365 |
| POO-H1 | Huagapo | -11.27 | -75.79 | Kanner et al. (2013) | 459 | 10.4 | 3800 |
| P09-H2 | Huagapo | -11.27 | -75.79 | Kanner et al. (2013) | 459 | 10.4 | 3800 |





| BOTO3 | Umajalanta–Chiflonkhakha | -18.12 | -65.77 | Apaéstegui et al. (2018) This study | 518 | 17.0 | 2650 |
|---|---|---|---|---|---|---|---|
| BOTO7 | Umajalanta–Chiflonkhakha | -18.12 | -65.77 | Apaéstegui et al. (2018) This study | 518 | 17.0 | 2650 |
| BOTO10 | Umajalanta–Chiflonkhakha | -18.12 | -65.77 | Apaéstegui et al. (2018) This study | 518 | 17.0 | 2650 |
| BOTO1 | Umajalanta–Chiflonkhakha | -18.12 | -65.77 | Apaéstegui et al. (2018) This study | 518 | 17.0 | 2650 |
| PAL3 | Palestina | -5.92 | -77.35 | Apaéstegui et al. (2014) | 1570 | 22.8 | 870 |
| PAL4 | Palestina | -5.92 | -77.35 | Apaéstegui et al. (2014) | 1570 | 22.8 | 870 |

**Table 1: Caves, locations and regional characteristics.**

### 3.2 Stalagmites

As presented in the original papers, all speleothems used in this study were collected with the initial goal

to reconstruct the SAMS using the $\delta^{18}O$ values. Therefore, stalagmites with a candle-type shape and uniform growth were preferentially collected in isolated chambers located far from the cave entrance. At such locations, temperature displays only minor variations throughout the year (characteristic of tropical caves), the air circulation is restricted, $CO_2$ concentrations are higher than atmospheric values, and the air is saturated in humidity. These conditions minimize the effects of ventilation, changes in temperature,

degassing and overall kinetic effects on the isotopic composition of the stalagmites.

### 3.3 Methods

To establish the correlation between the $\delta^{13}C$ and the growth rate from speleothems, we calculate the growth rate based on the length of the interval between two consecutive U/Th ages and calculate the mean $\delta^{13}C$ for the respective interval. For the correlation between the $\delta^{13}C$ values and local temperature and precipitation

we use a single average $\delta^{13}C$ value for each stalagmite, which was associated with the local mean temperature and precipitation reported in the original papers.

In order to assess the shared variance among the $\delta^{13}C$ cave sites, while considering the stalagmites' age uncertainties, we employ a Monte-Carlo Principal Component Analysis (MC-PCA), following the methodology of Campos et al. (2019) and Deininger et al. (2017). The MC-PCA results in a set of loading

patterns representing the $\delta^{13}C$ spatial variability, and corresponding Principal Components, which characterize the temporal variability of the loading patterns, each explaining a percentage of the shared variance. 1000 Monte-Carlo simulations were performed using the isotopic time series obtained from the interpolation of each individual stalagmite's age model. Isotopic records from the same cave or karst system were merged in a single time series. To combine records, a normalization (z-score) of the shared period

using the mean and standard deviation of the longer time series was applied, before reconstructing the time

period through the inverse operation. This procedure resulted in a set of eight time series, each corresponding to a site shown in Fig. 1. Four stalagmites presented in Fig. 1 (MV3, FN1, TRA7, BTV21a) were not included in the PCA because these records do not cover the entire period of the last 2000 years and no other stalagmites from the same karst systems exist that could be merged, or because their data

resolution is significantly lower (more than 15 years between individual data points in the geochronological model). Given the time span covered by most records, the MC-PCA was evaluated only for the period from 650 to 1950 CE. To avoid biased results induced by large differences in altitude and temperature, linear regression analyses between the $\delta^{13}$C data and temperature and altitude were performed excluding the sites from the high-altitude Andes (Umajalanta–Chiflonkhakha and Huagapo cave systems).

**4 Regional setting**

The speleothem $\delta^{13}$C records used here are distributed throughout tropical South America (Fig. 1), covering a region spanning the latitudes 4 °S to 21 °S and longitudes 42 °W to 76 °W. The domain comprises the following climates (according to the Köppen climate classification): monsoon (Am), tropical savanna (Aw), warm semi-arid (BSh) and humid subtropical (Cwa). The annual precipitation amount ranges from 450 to

2400 mm, mainly related with the SAMS and its subcomponent the SACZ (Novello et al. 2018). The equatorial portion is also under direct influence of the Intertropical Convergence Zone (ITCZ). Except for the sites located in the Andes, all records are located at altitudes below 700 m.a.s.l. (**Table 1**). At these lowland sites, precipitation amount and the length of the rainy season define the three main vegetation types (Fig. 1). High precipitation amounts that are well distributed over the year are typical of Tropical Forest,

while Caatinga is characterized by an environment with lower precipitation, concentrated during a few months. Cerrado, the main biome of central South America, present hydrological conditions in-between these two end-members. In general, temperature decreases with increasing latitude and altitude; the mean annual temperature in the lowlands ranges from 21.4 to 26.8 °C, while the sites in the Andes reach temperatures around 10 °C (**Table 1**).

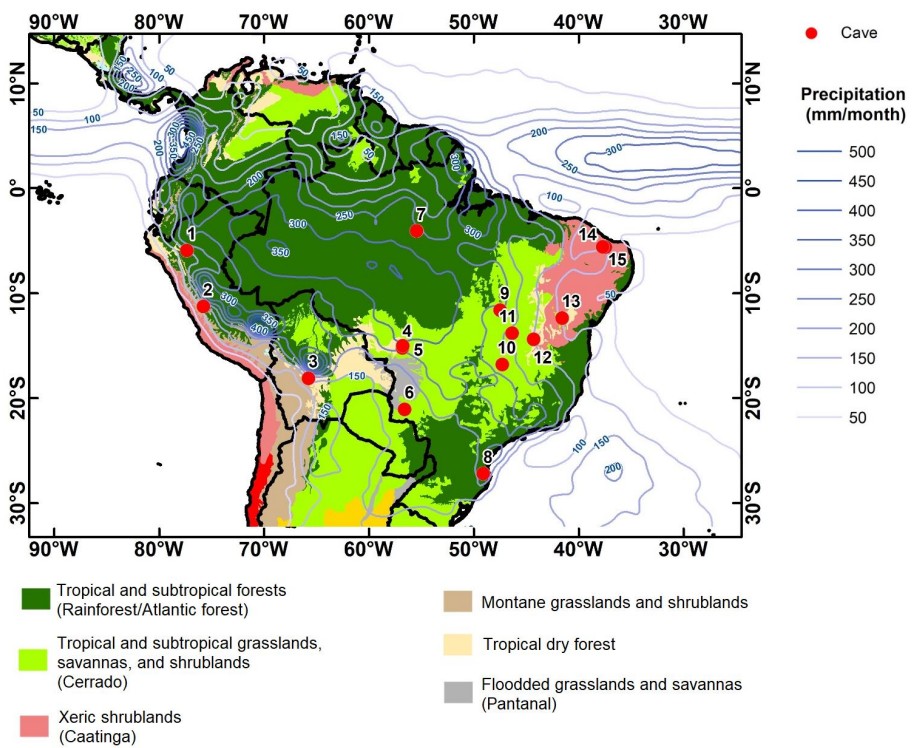

**Figure 1: Map of tropical South America with vegetation types (Olson et al., 2001), the main biomes, precipitation (blue isolines - mm/month, derived from the annual mean for the period from 1998 to 2017, with data from TRMM 3b43 – Huffman et al., 2014) and location of the study sites from the $\delta^{13}$C_2k_SA dataset. 1- Huagapo cave; 2- Palestina cave; 3- Umajalanta–Chiflonkhakha; 4- Pau d'Alho cave; 5- Curupira cave; 6- Járaguá cave; 7- Paraiso cave; 8- Botuverá cave; 9- Mata Virgem cave; 10- Tamboril cave; 11- São Matheus and São Bernardo cave system; 12- Anjos cave; 13- Diva, Torrinha, Lapa Doce caves; 14- Furna Nova cave; 15- Trapiá cave.**

## 5 Results

The $\delta^{13}$C values from the $\delta^{13}$C_2k_SA dataset ranges from -11.5 to 6.8 ‰, although the large majority of the data lie within -11 and -1 ‰ (Appendix A, Fig. A1). The highest values and largest variability stem from the TR5 stalagmite, ranging from -3.8 to 6.8 ‰ (amplitude of 10 ‰), which contrasts with the average amplitude from the other stalagmites of 4.5 ‰. The vegetation domains of Tropical Forests (Rainforest/Atlantic forest) and Cerrado include the speleothems with the lowest $\delta^{13}$C values (mean of -8.9 ‰ and -8.5 ‰, respectively), while the speleothems collected under the Caatinga domain have higher values (mean of -4.9 ‰) distributed over a large range (between 7.7 -4.9 ‰ and 3.1 ‰). High altitudes have the highest mean $\delta^{13}$C values of ~ -3.5 ‰ (Appendix A, Fig. A2)

The coefficient of determination (R-square) between the $\delta^{18}O$ and $\delta^{13}C$ values of each stalagmite (Hendy test, Hendy et al., 1971) is shown in Appendix A, Table A1. The highest coefficient of determination ($r^2$: 0.96, p<0.01) is displayed by the TR5 stalagmite, while the average $r^2$ of the $\delta^{13}C\_2k\_SA$ dataset is 0.29. Average $\delta^{13}C$ values from the speleothems display a high negative correlation with the annual mean precipitation amount ($r^2$: 0.67, p < 0.01) and a weaker, positive correlation with temperature ($r^2$: 0.45, p<0.05) of their respective regions (Fig. 2). No significant correlation was found with altitude (p>0.05). The relationships between the growth rate and average $\delta^{13}C$ values are statistically significant (p≤0.05) only in the stalagmites JAR4 ($r^2$: 0.21), DV2 ($r^2$: 0.50), TR5 ($r^2$: 0.75), LD12 ($r^2$: 0.70) and TRA7 ($r^2$: 0.20), all with negative slopes (Appendix A, Table A2).

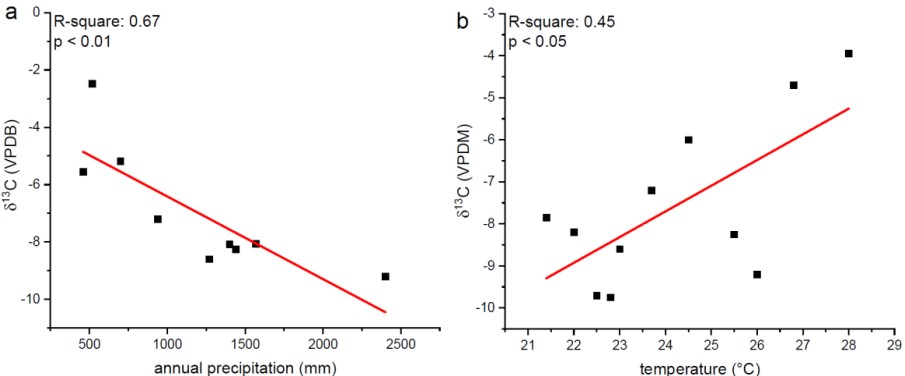

**Figure 2: Relationship between the $\delta^{13}C$ from the $\delta^{13}C\_2k\_SA$ dataset with annual precipitation (a) and annual mean temperature (b) of each study site.**

The MC-PCA methodology was applied to the full $\delta^{13}C\_2k\_SA$ dataset. The first principal component (PC1) explained ∼36 % of the total variance (Fig. 3 and 4). Positive loadings are associated with the $\delta^{13}C$ records from the caves Huagapo (Hua), Palestina (PAL), Jaraguá (Jar), and the merged records from the caves Pau d'Alho/Curupira (Alh) and São Bernardo/São Matheus (SBE), whereas negative loadings are linked to the records from the caves Diva de Maura/Torrinha/Lapa Doce (DV2), Paraiso caves (PAR) and Umajalanta–Chiflonkhakha (BOTO) (Fig. 3), indicating out of phase $\delta^{13}C$ variability between the two cave site groups. Positive values characterize the time interval of ∼600–1500 CE in PC1, while negative values are predominant during the period ∼1500–1950 CE, with an incursion to more negative values centered at ∼1180 CE (Fig. 4). PC2 explained ∼16 % of the variance, with large positive loadings displayed by the

records located in central Brazil, especially the TM0 and DV2 records, whereas large negative loadings are displayed by the records from high altitudes (PAL and BOTO). The other study sites present low and variable contributions for PC2. The temporal evolution of PC2 is characterized by predominantly positive values during the intervals 840-1110 CE and 1660-1950 CE, and negative values in the intervals 730-840

CE and 1110-1660 CE (Appendix A, Fig. A3).

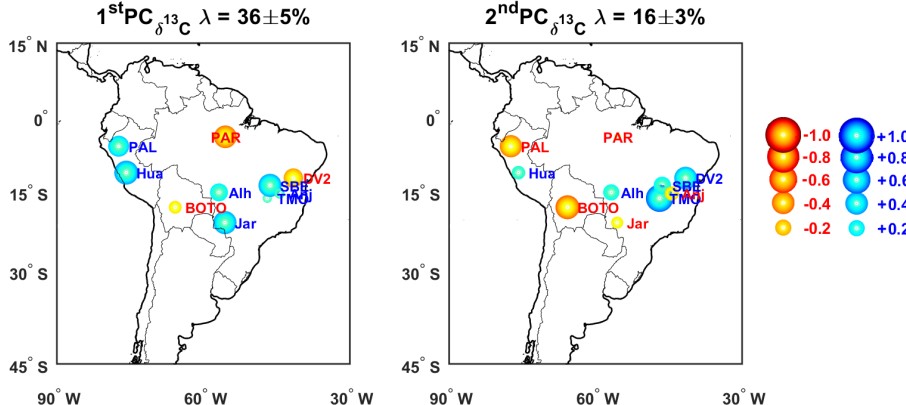

**Figure 3: Maps of South America with the main loadings of the Principal Component Analysis (PCA) and explained total variance. Blue and red dots represent positive and negative loadings, respectively. The magnitude of the loadings is represented by the size of the dots. The larger the dot, the more representative it's loading is**
**of the respective PC.**

### 6 Discussion

The host-rocks of the caves in this study are predominantly of Neoproterozoic age, located in the Brazilian shield (Appendix A, Table A1). Isotopic studies carried out in these carbonate rock sequences show a wide

range of values, varying between ~ -5 to 16 ‰ within the same rock unit or even the same outcrop (Appendix A, Table A1). The absence of isotopic studies in the same bedrock where the caves are located, precludes a precise quantification of the influence of these different rock sequences on the $\delta^{13}C$ values of the stalagmites. Of the study sites presented here, the $\delta^{13}C$ from its host-rock was only measured at Jaraguá and Trapiá caves (Novello et al., 2019; Utida et al., 2020). The dolomite of Jaraguá cave has $\delta^{13}C$ values of

~1 ‰, which is similar to the values documented in the stalagmites of this cave growing during the last glacial period, in contrast to the low values (~-8.6 ‰) documented for stalagmites covering the Holocene period. Novello et al. (2019) associated the more depleted Holocene carbon isotopic values from this cave



with increased soil thickness, denser vegetation and the establishment of vegetation dominated by $C_3$ plants above the cave. The same interpretation was adopted for the $\delta^{13}C$ record from Trapiá cave, located in the

northeast of Brazil (Utida et al., 2020). At this site, however, the climate and environmental conditions were out of phase with Jaraguá Cave during the Holocene and glacial periods, and high $\delta^{13}C$ values in the stalagmites, similar to bedrock values (~0 ‰), were documented during the Holocene. In summary, both regions experienced a larger isotopic contribution from host rocks to the $\delta^{13}C$ values of their stalagmites during periods of sparse vegetation and thin soil layers above the caves. Indeed, nowadays these are the

typical characteristics of the sites Huagapo, Umajalanta–Chiflonkhakha, Mata Virgem, Trapiá and Torrinha; all featuring stalagmites with high $\delta^{13}C$ values (Appendix A, Fig. A1 and Table A1), these sites are also located in elevated altitudes or under Caatinga domain. Aside from the stalagmites of these sites and ALHO6, all other stalagmites have $\delta^{13}C$ values lower than -6 ‰, indicating the predominance of thick soil with higher organic matter content from the regions of Cerrado and/or Tropical Forest, sourced from

vegetation formed by $C_3$ plants over the last 2000 years.

TR5 presents $\delta^{13}C$ values that are significantly higher than all other samples, and the abrupt increase in $\delta^{13}C$ occurs just before the complete stop of carbonate deposition at ~1920 CE and ~2006 CE (Appendix A, Fig. A1), both of which are periods of predominantly dry conditions in the region (Novello et al., 2012). High values followed by an abrupt decrease also occur after the depositional hiatus at ~1960 CE. The $r^2$ between

$\delta^{13}C$ and $\delta^{18}O$ is close to 1 (Appendix A, Table A1), which indicates that both isotopes underwent fractionation together in this sample. These results support the notion that the TR5 stalagmite was precipitated under strong kinetic effects due to high evaporation rates and/or significantly reduced dripping rates (long time for isotope re-equilibration between solution and cave atmosphere) at the moment of carbonate precipitation, which is accentuated during drier periods, culminating in a complete halt of

carbonate precipitation. Although the kinetic effect might be the main factor responsible for the high amplitude and isotopic values in TR5, this stalagmite still preserves paleoclimate information regarding the $\delta^{18}O$ in precipitation that co-varies with the $\delta^{18}O$ from other stalagmites in the region, as well as with pluviometric data from meteorological stations (Novello et al., 2012).

Most of the stalagmites from the $\delta^{13}C\_2k\_SA$ dataset feature coefficients of determination between their

$\delta^{13}C$ and $\delta^{18}O$ values ranging from 0.20 to 0.56 (Appendix A, Table A1). The $\delta^{18}O$ values in stalagmites from tropical South America covering the last 2000 years has been interpreted as a proxy of the SAMS (Azevedo et al., 2019; Novello et al., 2012, 2016, 2018; Apaéstegui et al., 2014; 2018; Wortham et al.,



2017; Vuille et al., 2012) and the SACZ, a continuous band of low-level wind convergence and precipitation that extends in a northwest-southeast direction across southeastern South America (Novello et al., 2018).

Since an increase in local rainfall promotes a decrease of PCP, an increase in the production of soil organic matter and favors $C_3$ plants over $C_4$ plants (all processes that result in low $\delta^{13}C$ values in stalagmites), we argue that a correlation between $\delta^{13}C$ and $\delta^{18}O$ values is the result of a close relationship between local hydroclimate and atmospheric convective processes (inferred by $\delta^{18}O$). This hypothesis is supported by the existing correlation between the annual rainfall amount and the average $\delta^{13}C$ values of the stalagmites ($r^2$:

0.67, Fig. 2), and by the PC1 loadings derived from the $\delta^{13}C\_2k\_SA$ dataset (Fig. 3a), showing a very similar spatial distribution as those displayed in PC1 derived from the $\delta^{18}O$ dataset from speleothems over tropical South America by Campos et al. (2019). Both PC1-$\delta^{13}C$ and PC1-$\delta^{18}O$ show positive loadings for the locations to the southwest of the SACZ and negatives loadings for those records located to the northeast (Fig. 3), which characterizes the SACZ precipitation dipole in South America (Campos et al., 2019; Novello

et al., 2018). The BOTO site (Fig. 3) is an exception in this scenario, but this region have partially being under the influence from a different climate system with a different moisture source in the past (Apaéstegui et al., 2018). This relationship between the $\delta^{13}C$ and local hydroclimate was already documented for the ALHO6 stalagmite using a multiproxy approach (Jaqueto et al., 2016).

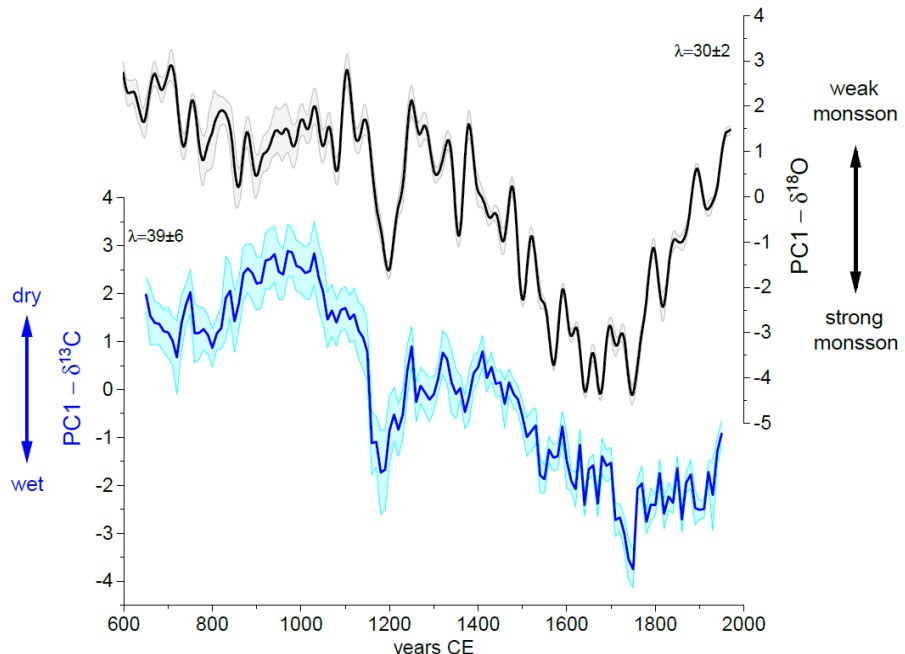


**Figure 4: Comparison between the first Principal component (PC1) derived from the δ¹³C_2k dataset (this study) representing the main mode of hydroclimate variability and PC1 based on the δ¹⁸O from stalagmites from South America (PC1 – δ¹⁸O) from Campos et al. (2019) representing the main mode of variability of the South American Monsoon System. The error propagation from the PC1s are show by colored outlines. λ**
**indicates the fraction of explained variance by each PC.**

Strong correspondence throughout the time-series between scores of PC1-δ¹³C and PC1-δ¹⁸O further

corroborates the overall coupling between the monsoon and local hydroclimate during the last ~1400 years

(Fig. 4). This coupling ceases after ~1750 CE, when the monsoon weakened at a faster rate than the local

hydrological response. However, the hydrologic variability inferred from PC1-δ¹³C is biased by vegetation

changes, which responds to other influences beyond the local rainfall amount, such as temperature and

atmospheric $CO_2$ (both of which are parameters that have increased significantly over the last 200 years).

PC2 explains only 16 % of the δ¹³C variability (Fig. 3, Appendix A, Fig. A3). It shows large positive

loadings at sites located in lowlands of central Brazil and negative loadings at sites located at the Andes

(Fig. 3). Since the δ¹³C_SA_2k dataset is significantly influenced by temperature (Fig. 2b), the small

contribution of PC2 to the δ¹³C_SA_2k dataset may be related to different effects of temperature at different

altitudes. Changes in continental temperature are amplified at higher elevation (Ohmura, 2012), leading to

a larger effect on the $\delta^{13}C$ values at sites located at higher altitudes, thereby increasing the isotopic differences between sites located at low and high elevation.

Growth rates also show a weak correlation with the $\delta^{13}C$ values from the $\delta^{13}C\_2k\_SA$ dataset. The correlation between the growth rate and $\delta^{13}C$ values is statistically significant (p-values $\leq 0.05$) only in the stalagmites JAR4, DV2, TR5, LD12 and TRA7. In these stalagmites, all regression coefficients present a negative slope, which is expected, as a higher growth rate leads to lower $\delta^{13}C$ values because the time for isotopic enrichment of the DIC through kinetic effects is reduced during carbonate precipitation

(Fohlmeister et al., 2020).

**7 Conclusions**

Here we present a new set of $\delta^{13}C$ records from speleothems collected over a broad region of tropical South America. These data were integrated with previously published speleothem $\delta^{13}C$ records to characterize the main controls on carbon isotope variations in this region. The predominance of $C_3$ plants above most of the

karst systems studies here is responsible for the low $\delta^{13}C$ values (< -6 ‰) in most of the speleothems, while local hydroclimate is the main driver behind its variability during the last two millennia. Unlike what was observed in the global compilation of $\delta^{13}C$ records for the period after 1900 CE (Fohlmeister et al., 2020), local temperature and growth rate play a minor role in the shaping the $\delta^{13}C$ values in the $\delta^{13}C\_2k\_SA$ dataset. The probable reason for this difference between studies is that most of the speleothems in our

database formed under tropical conditions, characterized by a limited temperature range, whereas the SISAL_v1 dataset studied by Fohlmeister et al. (2020) is biased towards records from high latitudes with a much large temperature range.

Using Monte Carlo Principal Component Analysis, we produce an index of the mean hydrologic conditions and its changes over tropical South America for the last two millennia, which is closely related to monsoon

variability for the period prior to 1750 CE. The recent break-down in the relationship between monsoon and local hydroclimate may have been caused by the increase in temperature and $CO_2$ during the current warm period; however, further studies are required to test this hypothesis.

**8 Data availability**

The new $\delta^{13}C$ records from $\delta^{13}C\_2k\_SA$ dataset can be found in Novello et al. (2020) available at

PANGAEA: (https://doi.pangaea.de/10.1594/PANGAEA.919050.



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

**Author contribution**: VFN designed the experiments, FWC and MV are the project PIs, JLPSC performed statistical analysis, NMS and JSM performed isotopic analyses, VA, GU, PJ, IK, LCRP and DOB assisted in the discussion and interpretations. AA created the map, GMPS provided a review on carbonate formations. VFN prepared the manuscript with contributions from all co-authors.


**Competing interests**: The authors declare that they have no conflict of interest.



**Acknowledgements**

We thank the São Paulo Research Foundation (FAPESP) for financial support through grants 2016/15807-

5 to VFN, 2017/50085-3, and 2019/07794-9 to FWC and the US National Science Foundation for award

OISE-1743738 to MV. We thank the Instituto Chico Mendes de Conservação da Biodiversidade (ICMBio)

for providing the permission (number 22424-8) to undertake the cave studies in Brazil.





**Appendix A**

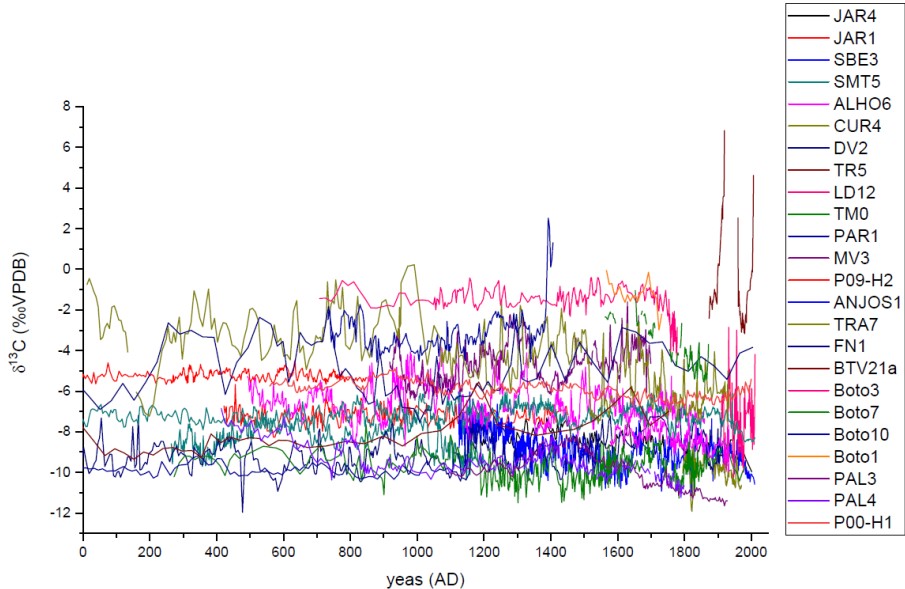

**Figure A1: δ¹³C records of stalagmites from tropical South America.**


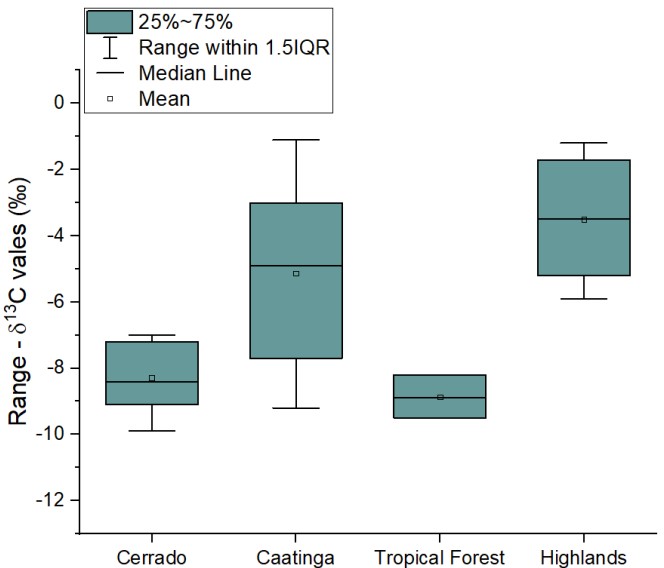

**Figure A2: Box plot with means of the δ¹³C values of stalagmites from each vegetation domain.**





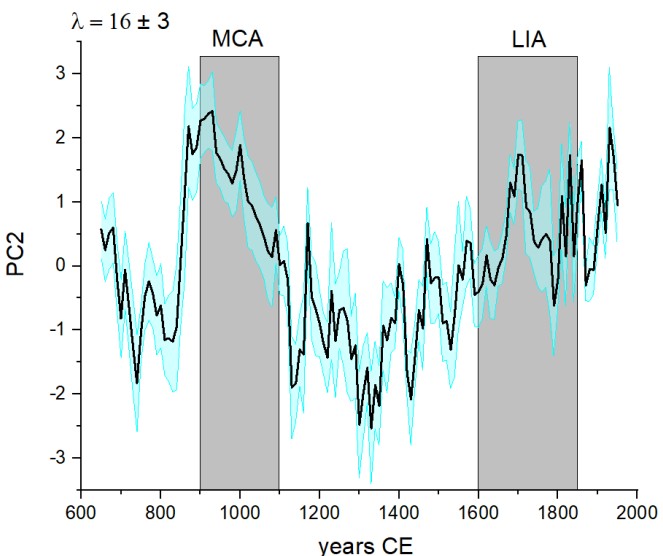

**Figure A3: PC2 derived from the δ¹³C_2k dataset, explaining 16±3 % of the total variance.**






| Stalagmite | δ18O mean | δ13C mean | R-square δ18O x δ13C | Time period covered (years BP) | Isotopic data points | Mean resolution (years) | δ13C of the main rock formation | Host geology |
|---|---|---|---|---|---|---|---|---|
| JAR4 | -4.8 | -8.5 | 0.50 (p<0.01) | -50 to 760 | 237 | 3.4 | 1* (1) | Bocaina Fm, Corumba Group (Neoproterozoic) |
| JAR1 | -3.8 | -7.2 | 0.21 (p<0.01) | 499 to 1508 | 318 | 3.2 | 1* (1) | Bocaina Fm Corumba Group (Neoproterozoic) |
| SBE3 | -4.3 | -8.9 | 0.04 (p<0.01) | -60 to 827 | 1710 | 0.5 | -5.5 to -3 (2) | Urucuia Fm, Mata da Corda Group (Cretaceous) |
| SMT5 | -4.2 | -8.3 | 0.13 (p<0.01) | 749 to 1686 | 575 | 1.6 | -5.5 to -3 (2) | Urucuia Fm, Mata da Corda Group (Cretaceous) |
| ALHO6 | -6.2 | -6.6 | 0.57 (p<0.01) | 90 to 1458 | 1169 | 1.2 | -5 to 0 (3) | Guia Fm, Araras Group (Neoproterozoic) |
| CUR4 | -7.6 | -9.9 | 0.25 (p<0.01) | -21 to 155 | 252 | 0.7 | -5 to 0(3) | Guia, Araras Group (Neoproterozoic) |
| DV2 | -3.7 | -9.2 | 0.35 (p<0.01) | 39 to 2765 | 538 | 5.0 | -9 to +10 (4) | Salitre Fm, Una Group (Neoproterozoic) |
| TR5 | -2.9 | -1.1 | 0.96 (p<0.01) | -57 to 76 | 90 | 1.5 | -9 to +10 (4) | Salitre Fm, Uma Group (Neoproterozoic) |
| LD12 | -3.2 | -7.7 | 0.35 (p<0.01) | -61 to 39 | 122 | 0.8 | -9 to +10 (4) | Salitre Fm, Una Group (Neoproterozoic) |
| TRA7 | -2.8 | -3.0 | 0.05 (p<0.01) | 17 to 5507 | 818 | 6.4 | 0* (5) | Jandaíra Fm, Apodi Group (Cretaceous) |
| FN1 | -2.3 | -4.9 | 0.87 (p<0.03) | -54 to 2275 | 88 | 26.5 | 0* (5) | Jandaíra Fm, Apodi Group (Cretaceous) |
| TM0 | -5.4 | -9.7 | 0.24 (p<0.01) | -32 to 1678 | 471 | 3.6 | -5 to +16 (6) | Sete Lagos Fm, Bambui Group (Neoproterozoic)) |
| ANJOS1 | -3.7 | -7.2 | 0.09 (p<0.01) | -57 to 2927 | 1272 | 2.4 | -5 t0 +2 (7) | Sete Lagoas Fm. Bambuí Group (Neoproterozoic) |
| BTV21a | -3.8 | -8.2 | 0.03 (p<0.01) | 196 to 9211 | 230 | 39.2 | X | Botuverá Fm, Brusque Group (Neoproterozoic) |
| PAR3 | -5.7 | -8.9 | 0.42 (p<0.01) | -48 to 768 | 144 | 5.7 | -2.2 to 5.2 (8) | Itaituba Fm Tapajos Group (Paleozoic) |
| PAR1 | -6.6 | -9.5 | 0.00 (p<0.44) | 714 to 4812 | 449 | 9.1 | -2.2 to 5.2 (8) | Itaituba Fm Tapajos Group (Paleozoic) |
| MV3 | -1.9 | -4.7 | 0.45 (p<0.01) | 250 to 1032 | 537 | 1.4 | X | Mato Virgem Fm, Natividade Group (Precambrian) |
| P00-H1 | -13.3 | -5.9 | 0.24 (p<0.01) | -50 to 1391 | 289 | 5.0 | X | Aramachay Fm, Pucara Group (Triassic) |
| P09-H2 | -13.0 | -5.2 | 0.20 (p<0.01) | 1099 to 7146 | 1272 | 5.7 | X | Aramachay Fm, Pucara Group (Triassic) |
| Boto3 | -9.7 | -1.7 | 0.26 (p<0.01) | 171 to 1242 | 256 | 4.0 | X | Miraflores Fm, Pucara Group (Cretaceous) |
| Boto7 | -10.8 | -3.9 | 0.00 (p=0.6) | 78 to 388 | 97 | 1.8 | X | Miraflores Fm, Pucara Group (Cretaceous) |
| Boto10 | -9.7 | -3.1 | 0.34 (p<0.01) | 543 to 1217 | 207 | 3.3 | X | Miraflores Fm, Pucara Group (Cretaceous) |
| Boto1 | -10.2 | -1.2 | 0.41 (p<0.01) | 218 to 383 | 20 | 7.9 | X | Miraflores Fm, Pucara Group (Cretaceous) |
| PAL3 | -7.1 | -10.1 | 0.00 (p=0.2) | 21 to 850 | 192 | 4.3 | X | Chambara Fm, Pucara Group (Triassic_Jurassic) |
| PAL4 | -7.1 | -9.4 | 0.25 (p<0.01) | 151 to 1536 | 248 | 5.4 | X | Chambara Fm, Pucara Group (Triassic_Jurassic) |

**Table A1: Stalagmites and their respectively data information. (1) Novello et al. (2019); (2) Lima (2011); (3) Nogueira et al., 2006; (4) Caird et al. (2017); (5) Utida et al., 2020; (6) Alvarenga et al., 2014; (7) Caetano-Filho et al., 2019; (8) Campos (2013). (*) Site where the δ13C values were measured in the bedrock that hosts the cave.**





| Stalagmite | Coefficient of determination between growth rate & $\delta^{13}C$ ($r^2$) | N° of segments |
|---|---|---|
| **JAR4** | **(-) 0.21 (p=0.05)** | **18** |
| JAR1 | (+) 0.24 (p=0.17) | 9 |
| SBE3 | (+) 0.29 (p=0.12) | 10 |
| SMT5 | (-) 0.35 (p=0.60) | 3 |
| ALHO6 | (+) 0.03 (p=0.63) | 11 |
| CUR4 | (+) 0.00 (p=0.94) | 7 |
| **DV2** | **(-) 0.50 (p=0.02)** | **10** |
| **TR5** | **(-) 0.75 (p<0.01)** | **7** |
| **LD12** | **(-) 0.70 (p=0.04)** | **6** |
| **TRA7** | **(-)0.20 (=0.03)** | **23** |
| FN1 | X | 3 |
| TM0 | 0.00 (p=0.87) | 12 |
| ANJOS | 0.00 (p=0.72) | 26 |
| BTV21a | X | 3 |
| PAR3 | (-) 0.26 (p=0.24) | 7 |
| PAR1 | X | 3 |
| MV3 | (-) 0.46 (p=0.10) | 7 |
| P00-H1 | (+) 0.55 (p=0.26) | 4 |
| P09-H2 | (-) 0.35 (p=0.29) | 4 |
| **Boto3** | **(-) 0.94 (p=0.01)** | **8** |
| Boto7 | X | 2 |
| Boto10 | (+) 0.36 (p=0.40) | 3 |
| Boto1 | X | 1 |
| PAL3 | (+) 0.65 (p=0.10) | 4 |
| PAL4 | (+) 0.12 (p=0.27) | 11 |

**Table A2: Coefficients of determination ($r^2$) between growth rate, calculated based on the length interval between ages, and the mean $\delta^{13}C$ values of each interval. Correlations with confidence levels equal to or higher than 95% ($p \leq 0.05$) are shown in bold. The signal "+" and "-" denotes positive and negative slopes, respectively.**