# Peer review of "δ13C values in stalagmites from tropical South America for the last two millennia"

_Earth System Science Data, 2020_

## Referee Comment (RC1) · Anonymous Referee #1 · 27 Aug 2020

Review of manuscript: ESSD-2020-184 by Novello et al., about a data set publication entitled 'd13C values in stalagmites from tropical South America for the last two millennia'

The authors present a data set of d13C values for several speleothems from South America. Some records in the data set were already earlier published, some are new. But in all cases the according d18O values were published elsewhere. The time interval of provided data in the new data set are strictly limited to the last 2ka, even when the stalagmites started their growth earlier. Furthermore, the authors provide some interpretation on the d13 C data sets, by applying correlation methods (with T and prcp) and principal component analysis. They conclude that the d13C values in speleothems from South America reflect changes in hydrology, which is closely related to changes

in monsoon variability.

I like the approach about focusing on stable C isotopes in speleothem, as this is an underrepresented field, as this isotope system is usually much more difficult to interpret. Unfortunately, I think the paper is not ready to be accepted in ESSD. In fact, I suggest to make major edits to the data base and the manuscript text. Please find my suggestions below. I hope they are of some help to improve this interesting work.

First, I want to evaluate the data set. To my knowledge, this data set indeed seems to include all speleothems, which are published so far from this region and grew during the last 2ka. From this point of view, the data set seems to be complete. I, however, do not understand, why the authors decided to publish only the d13C values for those parts of the speleothems covering the last 2 ka - even when some speleothems showed some growth in earlier periods. I understand that the interpretation is only focused on the last 2ka, but this shouldn't be a reason to not publish the older portions of the speleothem data (>2ka). In fact, my understanding of this journals scope is its interest in the data sets and their description and not so much on interpretation of data. For me it makes no sense to publish only parts of the available d13C data for some speleothems. I agree, that the data base must not contain data from additional South American speleothems, which grew exclusively earlier than 2 ka BP, but I would expect to find at least the full data sets of those speleothems, which are already included.

Another reason, why the d13C values of the older parts of the speleothems should be included, is the usefulness of the data set. I think, you intend to publish this data set with wanting the data to be used. But why should anyone use incomplete data and instead has to search for the full data set elsewhere?

In addition, I would also favor, if you could add the according d18O values to this data set. I know that they are already published in individual papers (but this is also true for some of the d13C records) and even in SISAL (see Comas-Bru et al., 2020, ESSD), but it would make things much easier, when trying to use the data, as most users are

also interested in d18O. And it would be most inconvenient, to extract the d18O data from SISAL and d13C data from this data set (and maybe even to search for the older parts of the d13C data sets elsewhere) to obtain both time series.

Otherwise the data are well stored and easy to access in Pangea. I like this. However, as this journal is more about data sets instead of interpretation, it should at least contain a section in the manuscript, which describes how to access the data (even, when the section will be short, as it is very easy to access them). You might write about the file format, which program is suited to access the data. What other parameters are provided. What are the metadata? How were the age models constructed? (Describe in more detail how you constructed the composites if you decide to include them in your data base (see below).)

Furthermore, in the data set itself, you should:

* check if all the d13C values are really reported against the PDB standard as stated in the column header. (I think, they might be measured against the VPDB standard, aren't they?)

* remove the 'age (ka BP)' column. First this column is redundant and second it is not helpful at all, as you provide only two digits. This is very unfortunate for your sub-decadal, high resolution records.

* provide more digits for the 'depth' column. Otherwise this column is also only of limited help.

* add information where to find the according d18O values (if you decide against including them here)

*add information, where to find the original U-Th data sets. (Those are important for researchers wanting to establish alternative age-depth models.)

* add the stacked records as well, which you constructed here. You are free to follow this advice or not, but to my opinion this would save a lot time for others, which want

to have such composite records and would make your data base much more attractive and unique. And you already have the composites at hand. So this shouldn't be much of an effort.

Second, I want to focus this review on the interpretation part of the data set. Line 60: Please be more precise and rephrase. Using a sentence like 'The initial source of C for speleothems is the soil $CO_2$ and tree roots' is very confusing. More correct would be that root respiration produces soil gas $CO_2$. But it is also produced by microbial decomposition of SOM. So, please rephrase.

Line 63: '... models were proposed to explain the dissolution ...' To my opinion, more correct would be '... models are the extreme cases of the dissolution ...', as there are a lot of intermediate conditions, under which carbonate can be dissolved and those are much more common.

Line 64: Please replace 'Initially' with 'In an open system'.

Line 66: You should finish the sentence by something like: '... fingerprint from the d13C composition of soil gas CO2.'

Line 69 to 71: You could be more precise, if you would add something like that the mixing ratio is expected to be approximately 1:1.

Line 72: You might should replace 'partially open system' by 'intermediate dissolution system'.

Line 74: I would not agree with using the <10% value anymore. In 2001, it was indeed the case that investigated speleothems showed 'usually' values like that. But, I wouldn't use this finding anymore, as since then, there were a lot of additional studies, showing that quite often the host rock contributes a lot more carbon to the speleothem than 10% (e.g. Jackson and McDermott, 2008; Hoffmann et al., 2010; Griffiths et al., 2012; Lechleitner et al., 2016a;2016b; Spötl et al., 2016; Bajo et al., 2017; some examples in Markowska et al., 2019; Therre et al., 2020).

Line 93 and 94: I have to admit, that I haven't read the paper by Mickler et al. (2019), but I wonder, if they really say that PCP removes preferentially 12C from the solution. Usually and according to published fractionation factors for C, during carbonate precipitation 13C is preferentially removed. Nevertheless, the d13C values will increase in the solution during PCP. That part of the sentence is correct. But this effect would be due to the simultaneous degassing of CO2. This process is responsible for an increase in 13C in DIC as the light C isotope is preferentially removed during degassing.

Line 95-96: '. . . exposure of seepage solution to air pockets along the epikarst flow routs . . .' Maybe that is also a reason, but usually, I thought, it is assumed that PCP is increased, due to larger drip intervals – leading to longer time intervals, where water is in contact with the cave atmosphere leading to more CO2 degassing and CaCO3 precipitation before the water reaches the speleothem top.

Line 98: Maybe it is more appropriate to talk about 'variations in PCP are climate related'. To my opinion, the occurrence of PCP itself is independent of climate, but its variations depend very well on climate conditions.

Line 174-175: I think it is appropriate to add 'seasonal' in this sentence as you refer to T-variations 'throughout the year' in the lines before.

Line 179-181: Here, you are describing how you calculate the average d13C data, which you use later (Fig. 2) to analyse them with respect to T and precipitation. I think you need to be a bit more precise here. To my understanding you calculate the mean d13C of the full record (for the last 2ka) and compare this with recent T. I doubt that this should be done. You might want to use the d13C values of the last 50 or 100 years or something like that. But using the mean d13C values of the last 2ka, shouldn't be compared with present-day T and precipitation - especially, after you mention a global T increase of 1.44 °C after 1850 CE (see line 145 of your manuscript).

Line 190: You describe here, how you constructed composite records. I understand from those lines, that you normalize the data before the stacking procedure starts. But

the stacking procedure itself is not described. Please elaborate more on this. Maybe it would also be a good idea to also present the stacked records in your data base (see above). This would be really a step forward.

Line 191: What do you mean by 'the inverse operation'?

Line 193: 'do not cover the entire period of the last 2000 years'. But you are calculating the PCA only from the period between 650-1950 CE (see line 197)? So, it shouldn't be a problem that those speleothems do not cover the full period of the last 2 ka. Or are they even shorter than this more limited period?

Line 197 to 199: I am not very familiar with the PCA, but my understanding of a PCA was, that it compares only relative changes in data, not absolute values. Thus, I do not understand your argument to exclude high-altitude sites due to large offsets. In addition, I do not understand, why you need a linear regression before you perform a PCA. Please explain.

Fig. 2: Indeed that are interesting relationships, however, I still wonder if you compare the mean d13C values calculated from the speleothems grown over the last 2ka with recent T and precipitation? Please clarify and recalculate the average of d13C over a more appropriate period (if not already done so). Furthermore, I do not fully understand why there are 9 dots in a) (d13C vs T) and 11 dots in b) (d13C vs prcp). Why both plots do not have the same number of data? In addition, according to Tab. 1 there exist data from 18 different caves. Please explain, why there are much less points in Fig. 2.

Line 282-284: I am not sure, if it is correct to argue that both regions experienced a larger isotopic contribution from host rocks to the d13C values. I think, there is much more what influences d13C than the host rock contribution. Such a claim could only be proved with additional 14C measurements. But usually other processes are more important for a change in d13C, as e.g., soil gas $CO_2$ production rate, fractionation processes in the cave. Both of those processes lead to elevated d13C values during drier conditions and should have a higher impact on d13C as changes in carbonate

dissolution processes.

Line 310-311: Maybe it is better to be more specific and add 'derived soil gas CO2' after 'soil organic matter', as this is more interesting for speleothem d13C than the amount of SOM alone.

Line 336-337: You argue about T and atmospheric CO2 to be responsible for the missing coupling between d13C and d18O after ~1750 CE. Around 1200 CE, there is a similar decrease in the PC1 of d18O and d13C. While d18O goes back to values similar to before this events, d13C does not. So there appears to be a similar decoupling of both isotope system as well. However, there is nothing comparable like T or CO2 increase. Thus, I wonder, how the decoupling at this period can be explained and if T and CO2 are indeed the most likely explanation for the decoupling after 1750CE. I do not have another explanation myself, but this observation leave me back in some doubt about the T and CO2 argument. That is really puzzling.

Line 340-341: You argue, that d13C is influenced by temperature here and refer to Fig. 2. But to my opinion, this plot shows only that the level of d13C values appear to depend on T (but still, you have to answer for which period you have calculated the average d13C values). I think, this is not a proof that the time series for d13C will also react on temperature in the same way. Therefore, please rephrase this sentence or leave it away, as you do not really need this.

Technical corrections:

Line 90: Please pay attentions to the spelling of 'Schubert and Jahren'.

Line 311: Replace 'low' by 'decreasing'

Line 330: You are using 'fraction' of explained variance, which calls for numbers between 0 and 1, but in Fig. 4 you provide numbers larger than 1. Please fix this.

Line 362: 'much larger'

---

## Referee Comment (RC2) · Anonymous Referee #2 · 5 Sep 2020

This paper presents 10 new speleothem $\partial$13C records included in a new dataset of 25 speleothem $\partial$13C records meant to characterize the last 2ka (in fact the last 1.4 ka) over tropical South America. This series of data is used to reconstruct a general pattern of the climate evolution during the medieval climate anomaly and the little ice age. Main results show that low $\partial$13C values are related to high C3 plant density in the soil above the cave and highlight a breakdown between monsoon variability and local hydroclimate after 1750 CE based on the establishment of an index associated to mean hydrologic conditions.

Main focus of the paper was to test the influence of local hydroclimate, temperature and changing vegetation types on the carbon isotopes. The use of C stable isotope in speleothems has been poorly explored with complex interactions among the dif-

ferent drivers which makes difficult the climatic or environmental interpretations and I acknowledge the attempt of the authors. However none of these drivers (temperature, hydroclimate and vegetation cover) is really discussed for each record and general conclusions are often based on specific observations without in-depth argumentation. I feel concerned by the fact that if temperature inluence is eventually discarded, there is little evidence of what would be the expression of a local hydroclimate and the associated vegetation types. Indeed no information was given by the authors about the vegetation cover that is supposedly at the origin of the $\partial$13C and the absence of calibration makes the demonstration rather poor. When I started to look for such information through the original publications, I found that vegetation was defined at only 3 records upon 25 (I had no access to Utida et al 2020) mixing up biome with composition of the vegetation type or physiognomy. For example the record of Tamboril is located within a Âń Native semi-deciduous forests. . .Âż, Paraíso is Âń densely covered by rainforest Âż and Mata Virgem is Âń located in the Eastern regions of the tropical Savannah known as 'Cerrado' and the Amazon Forest Âż. For the last one, it is impossible to know whether the Cerrado or the Amazon forest is characterizing the area. These are two different biomes, and the reader would like to know about the vegetation type that grows above the cave, as for instance gallery forest, grassland with Cactaceae, campo limpo. . .etc).

Under such poor descriptions of the vegetation cover, I have difficulties to believe the conclusion Âń Most locations were dominated by C3 plants over the last two millennia Âż. Finally the authors concluded that the dataset was able to show that $\delta$18O and $\delta$13C generally co-varied except at Tamboril cave and did not discuss the soil richness which was first expressed as a significant factor for $\partial$13C. Neither was discussed the composition of the local/regional vegetation types that grow today above the cave. Information are mixed up and some main points presented at the beginning of the ms are simply abandonned when reaching the discussion part. For instance, at Jaragua cave a decrease of 9 ‰ in the $\delta$13C values was interpreted as Âń resulting from a combination of changes above the cave, including: changes in the predominant vegetation type from C4 to C3, increase of organic matter and soil horizons Âż which does not

bring any strong information about the results. Moreover the breakdown between $\partial 18O$ and $\partial 13C$ after 1750 CE could also be due to deforestation and/or high fire activity. A comprehensive bibliography about the vegetation change and fire history of the different study areas is missing for the discussion. This paper needs a complete revision adding more precise information to support the interpretations and the conclusions. A proper discussion would also include the description and calibration of the vegetation and their influence on the obtained results. For these reasons I do not recommend this ms for publication.

Specific issues and comments

-Line 60 : importance of the tree roots : what about the herbs ?

-Line 86 what about the CAM plants which are abundant in some regions covered by the data set ?

-Line 127 last glacial

-Most of the sites located in the Caatinga show high abundance of Cactaceae which are C3 plants but do not represent dense forest neither rich organic soils.This should be mentionned and taken into account in the discussion.

-Comparisons with $\partial 18O$ records show that there is little difference with the $\partial 13C$ records regarding the climate interpretation and the authors come to the same conclusions except for the last centuries. Consequently, the gain of the $\partial 13C$ analyses in speleothem records is not clear.

-Some interpretation/conclusion sound fuzzy. For instance at Jaragua cave a decrease of 9 ‰ in the $\delta 13C$ values was first interpreted as Âń resulting from a combination of changes above the cave, including changes in the predominant vegetation type from C4 to C3, increase of organic matter and soil horizons Âż. Later in the paper the vegetation hypothesis is abandonned and the decrease was finally related to temperature and atmospheric CO2.

-How do we know that Âń Most locations were dominated by C3 plants over the last two millennia and are characterized by speleothem $\delta$13C values more depleted than -6 ‰ Âż as no description of the vegetation is given? Also Line 87 Âń Variations in soil $\delta$13C values and their evolution over time are controlled by carbon inputs from vegetation, which is proportional to the organic matter amount and vegetation density; Âż was promising but without any description of the vegetation it is difficult to evaluate the influence of the plant composition on the $\partial$13C.

-Line 209-213 Cerrado is also a tropical forest so what do you mean by Âń Tropical Forest Âż ? also in the legend of figure 1 Rainforest/Atlantic forest, Atlantic forest is a biome that includes many vegetation types among them the rainforest (with no majuscule). What is the legend referring to ? Also line 289 what do the authors mean by Âń a tropical forest Âż ? is it a rainforest or a seasonal forest or a dry forest or a semi-deciduous forest or a cloud forest?

-Line 211 the 3 sites that are called Âń andean sites Âż are located in very different environmental conditions : Huagapo, 3850 m asl should correspond to a high elevation grassland as the tree line never climbed further up than $\sim$3500 m in the last 3000 years, Umajalanta 2650 m asl Âń monsoon-related convection and condensation over the Amazon Basin Âż which corresponds to a cloud forest, Palestina, 870 m asl, the transition between rainforest to cloud forest.

-Line 228-230 Are you saying that Cerrados and rainforest have the same $\partial$13C ? Âń The vegetation domains of Tropical Forests (Rainforest/Atlantic forest) and Cerrado include the speleothems with the lowest $\delta$13C values (mean of -8.9 ‰ and -8.5 ‰ respectively),.. Âż

-Line 232 Appendix Fig A2 What is meant by highland records ? they are located in three different vegetation covers (see above). Then the $\partial$13C should be different for each of these vegetation types. Why isn't it so ?

-Line 235-238 then it is not correlated with vegetation as it was explained at the beginning of the manuscript?

-Line 248 etc MCP-PCA could also include the degree of opening of the vegetation types that grow above the cave in the discussion.

-Line 254 this could have been shown also with $\partial 18O$ ?

-Line 278 denser vegetation : this could be checked

-Line 290 I disagree here because in the Caatinga there is no C4 plants, a high density of C3 deciduous trees plus the Cactaceae (C3 carbon metabolism).

-Line 291 TR5 add the vegetation type/ the environment of the cave

-Line 291-303 : the $\partial 13C$ relates to climate and evaporation ? not anymore with soil thickness ? and what about plant assemblages ? these last two points are not discussed (also line 333) when they were listed as main factors of $\partial 13C$ variability earlier in the ms.

-Line 311 C3 plants do show a broad range of physiognomies in the Tropics Line 311 : this was not really demonstrated in the ms neither in Novello et al 2019

-Line 313 If $\partial 13C$ reflects hydroclimate, why analysing the Carbon stable isotope as $\partial 18O$ proved to be efficient in that field?

-Line 334 after 1750 CE, may be consider deforestation ? which could eventually introduce a discussion on the vegetation cover ?

Line 336 which vegetation changes and where ?

Line 337 vegetation responds to other influence...yes I thought it was soil thickness ? Although here it is the temperature that is inferred.

Line 349 so again not related to vegetation

Line 354 This could be an important issue to consider for instance when comparing with other compilation

Line 354 356 I do not agree with this conclusion. The predominance of C3 plants on the study sites was not shown.

Line 363 365 what is the novelty here ? it was already showed by $\partial 18O$

Line 366-367 and deforestation ?

Tables and Figures

Table 1 Add at least the biome in front of each site and the vegetation type that grows above the cave.

Table A1 add the length of the speleothem

Figure 1 Legend figure 1 : Cerrados with S

Figure 2 what don't we see 25 points ?

Figure A1 is not lisible

Figure A2 : what do you call a vegetation domain ? a biome ? a vegetation type ? anyway this is none of the category represented here. It is not possible to relate the $\partial 13C$ to the vegetation cover. This point needs further development in your argumentation.

---

## Author Comment (AC1) · 12 Oct 2020

Response to reviewer 1. We wish to thank the reviewer for accepting to review our manuscript and provide such a careful and detailed review. There is a clear and genuine interest by the reviewer to help improve our manuscript, for which we are most grateful. We believe that all topics were addressed in the revision and clarified by the comments below.

reviewer comment (RC)

RC: Review of manuscript: ESSD-2020-184 by Novello et al., about a data set publication entitled 'd13C values in stalagmites from tropical South America for the last two millennia' The authors present a data set of d13C values for several speleothems from

South America. Some records in the data set were already earlier published, some are new. But in all cases the according d18O values were published elsewhere. The time interval of provided data in the new data set are strictly limited to the last 2ka, even when the stalagmites started their growth earlier.

Answer: We did actually provide the entire stalagmite records, including data prior to 2ka. Please see the ages and data of the stalagmites DV2, ANJOS1, TRA7, BTV21a and FN1 that have data older than 2ka.

RC: Furthermore, the authors provide some interpretation on the d13 C data sets, by applying correlation methods (with T and prcp) and principal component analysis. They conclude that the d13C values in speleothems from South America reflect changes in hydrology, which is closely related to changes in monsoon variability. I like the approach about focusing on stable C isotopes in speleothem, as this is an underrepresented field, as this isotope system is usually much more difficult to interpret. Unfortunately, I think the paper is not ready to be accepted in ESSD. In fact, I suggest to make major edits to the data base and the manuscript text. Please find my suggestions below. I hope they are of some help to improve this interesting work.

Answer: Yes, thank you – the comments were most definitely helpful.

RC: First, I want to evaluate the data set. To my knowledge, this data set indeed seems to include all speleothems, which are published so far this region and grew during the last 2ka. From this point of view, the data set seems to be complete. I, however, do not understand, why the authors decided to publish only the d13C values for those parts of the speleothems covering the last 2 ka - even when some speleothems showed some growth in earlier periods. I understand that the interpretation is only focused on the last 2ka, but this shouldn't be a reason to not publish the older portions of the speleothem data (>2ka). In fact, my understanding of this journals scope is its interest in the data sets and their description and not so much on interpretation of data. For me it makes no sense to publish only parts of the available d13C data for some

speleothems. I agree, that the data base must not contain data from additional South American speleothems, which grew exclusively earlier than 2 ka BP, but I would expect to find at least the full data sets of those speleothems, which are already included. Another reason, why the d13C values of the older parts of the speleothems should be included, is the usefulness of the data set. I think, you intend to publish this data set with wanting the data to be used. But why should anyone use incomplete data and instead has to search for the full data set elsewhere?

Answer: Please see our answer above related to this issue. We did provide the entire stalagmite records, including data prior to 2ka, when available. However, most of the stalagmites grew within the last 2ka.

RC: In addition, I would also favor, if you could add the according d18O values to this data set. I know that they are already published in individual papers (but this is also true for some of the d13C records) and even in SISAL (see Comas-Bru et al., 2020, ESSD), but it would make things much easier, when trying to use the data, as most users are also interested in d18O. And it would be most inconvenient, to extract the d18O data from SISAL and d13C data from this data set (and maybe even to search for the older parts of the d13C data sets elsewhere) to obtain both time series.

Answer: As the reviewer correctly pointed out, the d18O datasets are already published and available elsewhere with their respective DOI. We prefer to avoid conflicts with previous publications and potential issues with archiving the same data in different formats in multiple repositories. We therefore provide only the unpublished data, in this case the d13C, which is the subject of this paper. However, it may also be worthwhile pointing out, that the first author of this paper (Dr. Novello) serves as the coordinator of SISAL for the South American domain, and in this role, he is committed to providing the d13C data from this paper to the next version of SISAL database. Thereby, the speleothem community will have easy access to both isotopes, as well to all information regarding these stalagmites and caves through the SISAL database.

RC: Otherwise the data are well stored and easy to access in Pangea. I like this. However, as this journal is more about data sets instead of interpretation, it should at least contain a section in the manuscript, which describes how to access the data (even, when the section will be short, as it is very easy to access them). You might write about the file format, which program is suited to access the data. What other parameters are provided. What are the metadata? How were the age models constructed? (Describe in more detail how you constructed the composites if you decide to include them in your data base (see below).

Answer: As the reviewer stated, the data is stored and easily accessible via Pangaea. The direct link to access the data in Pangaea is provided in the paper. When this manuscript was submitted, the Pangaea webpage initially put restricted access to the data. However, we have contacted the editor of Pangaea and all restrictions to access the data have been removed. With a simple click on the link provided in the paper the reader will be taken to the dataset page. The data are available in the formats text and HTML (standard formats from PANGAEA) and no special software required to read them. The other parameters provided together with the d13C data are "depth' and "age", which are both intuitive and discussed in the manuscript. We do not supply any kind of metadata and the geochronology provided was taken from the original papers (focused on the d18O), as discussed in the manuscript. When standardizing the records for our statistical analyses, we performed new data interpolations between isotopes and ages, which are described in the section "Methods". Following the reviewer's suggestion, we improved this section and the description of our age models.

RC: Furthermore, in the data set itself, you should: * check if all the d13C values are really reported against the PDB standard as stated in the column header. (I think, they might be measured against the VPDB standard, aren't they?) Answer: Thank you for pointing this out. In fact, they were measured against VPDB. However, this error was introduced when the editor of Pangaea provided the label of these columns as PDB. We will contact them to change the label back to VPDB.

RC: * remove the 'age (ka BP)' column. First this column is redundant and second it is not helpful at all, as you provide only two digits. This is very unfortunate for your subdecadal, high resolution records.

Answer: The editor of Pangaea provided this column. In the original spreadsheet that we sent to the website only the column "age (years AD) was included". It appears, however, as if the "age (BP)" column" is a standard from Pangaea.

RC: * provide more digits for the 'depth' column. Otherwise this column is also only of limited help. Answer: One tenth of a mm is the limit of the precision when sampling stalagmites the conventional way. All stalagmites presented in this study were sampled with a distance between the datapoints $\geq$ 0.1 mm. Thus, more digits in this column have no meaning.

RC: * add information where to find the according d18O values (if you decide against including them here). *add information, where to find the original U-Th data sets. (Those are important for researchers wanting to establish alternative age-depth models.)

Answer: This information is included in section 3.1 of our manuscript, where we list the original sources of each record in Table 1.

RC: * add the stacked records as well, which you constructed here. You are free to follow this advice or not, but to my opinion this would save a lot time for others, which want to have such composite records and would make your data base much more attractive and unique. And you already have the composites at hand. So this shouldn't be much of an effort.

Answer: Thank you for this advice; we will provide these data as well.

RC: Second, I want to focus this review on the interpretation part of the data set. Line 60: Please be more precise and rephrase. Using a sentence like 'The initial source of C for speleothems is the soil CO2 and tree roots' is very confusing. More correct would be that root respiration produces soil gas CO2. But it is also produced by microbial

decomposition of SOM. So, please rephrase.

Answer: We agree. We have rephrased this sentence to "The initial source of carbon for speleothems is the CO2 present in the soil, mainly provided by plant roots' respiration and decomposition of organic matter".

RC: Line 63: ': : : models were proposed to explain the dissolution : : :' To my opinion, more correct would be ': : : models are the extreme cases of the dissolution : : :', as there are a lot of intermediate conditions, under which carbonate can be dissolved and those are much more common.

Answer: We agree. We rephrased this sentence and now write: "Open and closed system models were proposed to explain two extreme cases of dissolution of calcium carbonate in the percolating solution".

RC: Line 64: Please replace 'Initially' with 'In an open system'.

Answer: Done.

RC: Line 66: You should finish the sentence by something like: ': : : fingerprint from the d13C composition of soil gas CO2.'

Answer: Thank you for this suggestion. We rephrased the sentence to "…the bicarbonate in solution receives the $\delta$13C fingerprint of this reservoir".

RC: Line 69 to 71: You could be more precise, if you would add something like that the mixing ratio is expected to be approximately 1:1.

Answer: Between lines 69 and 71 we write; "The rock dissolution is limited by the initial amount of CO2 and, consequently, through this process the $\delta$13C from the bedrock influences the isotopic composition of the remaining solution". We believe that this statement clearly indicates that the bedrock only has an influence on the isotopic composition. It does not suggest that the mixing ratio is 1:1.

RC: Line 72: You might should replace 'partially open system' by 'intermediate dissolution system'.

Answer: We rephased the sentence to: "the interaction between the percolation solution and the host-rock occurs as an intermediate way between open and closed systems".

RC: Line 74: I would not agree with using the <10% value anymore. In 2001, it was indeed the case that investigated speleothems showed 'usually' values like that. But, I wouldn't use this finding anymore, as since then, there were a lot of additional studies, showing that quite often the host rock contributes a lot more carbon to the speleothem than 10% (e.g. Jackson and McDermott, 2008; Hoffmann et al., 2010; Griffiths et al., 2012; Lechleitner et al., 2016a;2016b; Spötl et al., 2016; Bajo et al., 2017; some examples in Markowska et al., 2019; Therre et al., 2020).

Answer: We agree. Thank you for indicating these references. We removed this last sentence from our manuscript, which in our opinion, does not impact the text.

RC: Line 93 and 94: I have to admit, that I haven't read the paper by Mickler et al. (2019), but I wonder, if they really say that PCP removes preferentially 12C from the solution. Usually and according to published fractionation factors for C, during carbonate precipitation 13C is preferentially removed. Nevertheless, the d13C values will increase in the solution during PCP. That part of the sentence is correct. But this effect would be due to the simultaneous degassing of CO2. This process is responsible for an increase in 13C in DIC as the light C isotope is preferentially removed during degassing.

Answer: Maybe this sentence was not clear. Indeed, the PCP does not preferentially remove the 12C and it was never our intention to imply that. We rephrased this sentence to avoid any misunderstanding. Now the sentence reads: "$\delta$13C values from dissolved inorganic carbon (DIC) can undergo fractionation through prior calcite precipitation (PCP), which is forced by CO2 degassing that preferentially removes 12C from the solution to the cave atmosphere (Mickler et al., 2019), depleting 12C from the

final isotopic product recorded in stalagmites (Baker et al., 1997)".

RC: Line 95-96: ': : : exposure of seepage solution to air pockets along the epikarst flow routs : : :' Maybe that is also a reason, but usually, I thought, it is assumed that PCP is increased, due to larger drip intervals – leading to longer time intervals, where water is in contact with the cave atmosphere leading to more CO2 degassing and CaCO3 precipitation before the water reaches the speleothem top.

Answer: Both are the same process. Since the solution is in contact with air, the CO2 degassing occurs, which increases the PCP. But the reviewer is correct to point out the importance of longer drip intervals. We rephrased the sentence to incorporate this suggestion. Now the sentence reads: "PCP increases during drier periods due to the increased exposure of seepage solution to air. This can occur during the contact of the solution with air pockets along the epikarst flow routes and/or with the increase of dripping interval, where the solution is exposed to the cave air for longer time at the stalactites; both results in CO2 degassing from the solution, promoting the carbonate precipitation in the epikarst and/or stalactites…"

RC: Line 98: Maybe it is more appropriate to talk about 'variations in PCP are climate related'. To my opinion, the occurrence of PCP itself is independent of climate, but its variations depend very well on climate conditions.

Answer: We agree. We reprhased this sentence to: "the variations in the PCP rate are climate related".

RC: Line 174-175: I think it is appropriate to add 'seasonal' in this sentence as you refer to T-variations 'throughout the year' in the lines before.

Answer: Done. Now the sentence reads: "These conditions minimize the seasonal effects of ventilation and temperature, degassing and overall kinetic effects on the isotopic composition of the stalagmites."

RC: Line 179-181: Here, you are describing how you calculate the average d13C data,

which you use later (Fig. 2) to analyse them with respect to T and precipitation. I think you need to be a bit more precise here. To my understanding you calculate the mean d13C of the full record (for the last 2ka) and compare this with recent T. I doubt that this should be done. You might want to use the d13C values of the last 50 or 100 years or something like that. But using the mean d13C values of the last 2ka, shouldn't be compared with present-day T and precipitation - especially, after you mention a global T increase of 1.44 _C after 1850 CE (see line 145 of your manuscript).

Answer: The reviewer understood correctly. To clarify this issue we improved this sentence that now reads: "For the correlation between the $\delta$13C values and local temperature and precipitation we use a single average $\delta$13C value for each stalagmite calculated over the last two millennia". However, we emphasize in the text that not all stalagmites cover the entire 2ka, and that we use the full time period that is common to most records. We understand that the comparison of the mean d13C values over the last 2ka with current temperature and precipitation may seem inappropriate. However, only few records have data available for the last 50 years, and some of the others have only few data points during this time interval. If we consider the last 100 years, most of the records will have data only for the period at the beginning of the XX century, which is not a representative time period. Thus, if we use only data from the recent period, we are limiting the use of our dataset and introducing a regional bias by excluding most of the records. The comparison over the full record allow us to use all stalagmites to provide a spatially more complete picture of the records from South America. Furthermore, with this approach, we minimize possible inconsistencies in the data related to geochronology and resolution. In theory, the recent d13C values should be more closely related to current temperature and precipitation. However, as we show in Figure 2, using the d13C average of the full record we still obtain a high R-square (0.67 for precipitation and 0.45 for temperature) with high statistical significance (p<0.01), which indicates that average d13C values over the last 2 ka are indeed representative of the current climate situation.

RC: Line 190: You describe here, how you constructed composite records. I understand from those lines, that you normalize the data before the stacking procedure starts. But the stacking procedure itself is not described. Please elaborate more on this. Maybe it would also be a good idea to also present the stacked records in your data base (see above). This would be really a step forward.

Answer: This methodology is not new; thus, we had provided the references of Deininger et al. (2017) and Campos et al. (2019) for this data treatment. But following the reviewer's suggestion we have improved our description in the new version of the manuscript. Please see our new description in the Methods section. We will also adopt the recommendation and provide the stacked records. Thank you for this suggestion.

RC: Line 191: What do you mean by 'the inverse operation'?

Answer: As stated above, we have improved the description of our methods. In the new version, this statement is no longer included.

RC: Line 193: 'do not cover the entire period of the last 2000 years'. But you are calculating the PCA only from the period between 650-1950 CE (see line 197)? So, it shouldn't be a problem that those speleothems do not cover the full period of the last 2 ka. Or are they even shorter than this more limited period?

Answer: They are indeed even shorter than this time period. See Table A1. We have rephrased this sentence to make this information clearer. Now the sentence reads: "Four stalagmites presented in Fig. 1 (MV3, FN1, TRA7, BTV21a) were not included in the PCA because these records cover only a limited time period and no other stalagmites from the same karst systems exist that could be merged..."

RC: Line 197 to 199: I am not very familiar with the PCA, but my understanding of a PCA was, that it compares only relative changes in data, not absolute values. Thus, I do not understand your argument to exclude high-altitude sites due to large offsets. In

addition, I do not understand, why you need a linear regression before you perform a PCA. Please explain.

Answer: The reviewer misunderstood. Nothing that we state in this sentence is related to the PCA. This confusion occurred because this sentence was included in the same paragraph where we are describing the PCA. To clarify this issue, we rephrased this sentence and moved it to a new paragraph in Methods section.

RC: Fig. 2: Indeed that are interesting relationships, however, I still wonder if you compare the mean d13C values calculated from the speleothems grown over the last 2ka with recent T and precipitation? Please clarify and recalculate the average of d13C over a more appropriate period (if not already done so).

Answer: See our comment above.

RC: Furthermore, I do not fully understand why there are 9 dots in a) (d13C vs T) and 11 dots in b) (d13C vs prcp). Why both plots do not have the same number of data? In addition, according to Tab. 1 there exist data from 18 different caves. Please explain, why there are much less points in Fig. 2.

Answer: These graphs were made using the d13C values listed in Table A1 e and precipitation and temperature values listed in Table 1. As can be seen in these tables, some of the stalagmites (from the same or different caves) are from regions with the same amount of precipitation and/or temperature, however, they have slightly different isotopic values. When this happens, we calculated the average between all isotopic values that are represented by the same amount of precipitation (or temperature), which resulted in the lower number of the data points. We use this approach to avoid a spatial bias, since some climatic regions have many more stalagmites (and caves) than others However, we noted that we did not adequately explain this aspect in the text. Now we provide this information in the methods section.

RC: Line 282-284: I am not sure, if it is correct to argue that both regions experienced

a larger isotopic contribution from host rocks to the d13C values. I think, there is much more what influences d13C than the host rock contribution. Such a claim could only be proved with additional 14C measurements. But usually other processes are more important for a change in d13C, as e.g., soil gas CO2 production rate, fractionation processes in the cave. Both of those processes lead to elevated d13C values during drier conditions and should have a higher impact on d13C as changes in carbonate dissolution processes.

Answer: We agree. We rephrased this sentence to: "... both regions experienced more positive $\delta$13C values in their stalagmites during periods of sparse vegetation and thin soil layers above the caves".

RC: Line 310-311: Maybe it is better to be more specific and add 'derived soil gas CO2' after 'soil organic matter', as this is more interesting for speleothem d13C than the amount of SOM alone.

Answer: Done.

RC: Line 336-337: You argue about T and atmospheric CO2 to be responsible for the missing coupling between d13C and d18O after _1750 CE. Around 1200 CE, there is a similar decrease in the PC1 of d18O and d13C. While d18O goes back to values similar to before this events, d13C does not. So there appears to be a similar decoupling of both isotope system as well. However, there is nothing comparable like T or CO2 increase. Thus, I wonder, how the decoupling at this period can be explained and if T and CO2 are indeed the most likely explanation for the decoupling after 1750CE. I do not have another explanation myself, but this observation leave me back in some doubt about the T and CO2 argument. That is really puzzling.

Answer: We do not agree that the decoupling between the PC1s during these two time periods are similar. Around 1200 CE, d18O goes back to values similar to before and d13C does not, but both PCAs present the same behavior (although with different range), however, after 1750 the behavior of both PCAs are clear different. The main

control of the d13C values in the stalagmites is the local hydrology that drives PCP and the CO2 production in soil, while the main control of the d18O values is the amount effect (an atmospheric process of the convection over the entire continent). We explain this on lines 91-109. Thus, the processes driving d13C and d18O variability are different, although related. Because that, differences between both PC1s are expected. We believe that this was case for the period around 1200.

RC: Line 340-341: You argue, that d13C is influenced by temperature here and refer to Fig. 2. But to my opinion, this plot shows only that the level of d13C values appear to depend on T (but still, you have to answer for which period you have calculated the average d13C values). I think, this is not a proof that the time series for d13C will also react on temperature in the same way. Therefore, please rephrase this sentence or leave it away, as you do not really need this.

Answer: We agree. We have removed this sentence.

RC: Technical corrections: Line 90: Please pay attentions to the spelling of 'Schubert and Jahren'.

Answer: We have corrected this error.

RC: Line 311: Replace 'low' by 'decreasing'

Answer: Done.

RC: Line 330: You are using 'fraction' of explained variance, which calls for numbers between 0 and 1, but in Fig. 4 you provide numbers larger than 1. Please fix this.

Answer: We replaced "fraction" with "percentage".

RC: Line 362: 'much larger'

Answer: We have corrected this error.

---

## Author Comment (AC2) · 12 Oct 2020

Response to the Reviewer 2

We appreciate the time and effort of the reviewer to evaluate our manuscript. We understand that this reviewer is not from the speleothem community, which we believe is a positive aspect, showing that our data can be of interest to a broad audience. Although we do not agree with some of the reviewer's comments, they have certainly helped us to clarify possible confusing points in our text, making her (his) contribution extremely important for our work. This reviewer appears to have been somewhat confused about what is our introduction and what is our discussion/conclusion section. For example, all comments provided below by the reviewer are related to vegetation. However, vegetation plays only a minor role in our interpretation. Our conclusion is that the d13C in the stalagmites from South America is related with local hydrology due to the process called Prior Calcite Precipitation (PCP), which occurs in the epikarst and is independent from vegetation. We mention the vegetation mainly in the introduction (section 1 and 2) as a part of the overall background information. To avoid this kind of misunderstanding we have removed and/or changed some sentences regarding vegetation that are not important to our discussion. We believe that all topics have been addressed in our revision and clarified by the comments below.

Reviewer Comment (RC)

RC: This paper presents 10 new speleothem d13C records included in a new dataset of 25 speleothem d13C records meant to characterize the last 2ka (in fact the last 1.4 ka) over tropical South America.

Answer: We present 13 new speleothem d13C records and the time period studied in the manuscript was the last 2 ka. Only the Principal Component Analysis (PCA) was performed over a time period of 1.4 ka because most of the records only cover this time period. We also wish to highlight that some of the stalagmites studied here are much older than 2 ka, and these data are also presented in the dataset that we are making available.

RC: This series of data is used to reconstruct a general pattern of the climate evolution during the medieval climate anomaly and the little ice age. Main results show that low d13C values are related to high C3 plant density in the soil above the cave and highlight a breakdown between monsoon variability and local hydroclimate after 1750 CE based on the establishment of an index associated to mean hydrologic conditions. Main focus of the paper was to test the influence of local hydroclimate, temperature and changing vegetation types on the carbon isotopes. The use of C stable isotope in speleothems has been poorly explored with complex interactions among the different drivers which makes difficult the climatic or environmental interpretations and I acknowledge the attempt of the authors. However none of these drivers (temperature, hydroclimate and vegetation cover) is really discussed for each record and general conclusions are often based on specific observations without in-depth argumentation.

Answer: As was mentioned by the reviewer, we discuss in our introduction that the d13C data from stalagmites have been ignored in many publications around the world, instead focusing only on the d18O values, mainly due to the complexity of d13C interpretations. The main aim of our study was therefore to provide a large set of speleothems d13C data to the community to encourage further research into the use and interpretation of this proxy. It was never our intention to discuss the drivers in each individual record. As we mention in sections 1 and 2, the d13C in stalagmites is sensitive to several drivers, which makes it difficult (and sometimes impossible) to disentangle competing processes. In fact, the complexity of d13C interpretations is seen as the main reason why these data have hitherto been neglected in many publications (as we mention in our introduction). But we believe there is value in presenting multiple records from the same continent and investigating their commonalities through a principal component analysis, thereby interpreting this proxy from a new perspective. Thus, this paper aims to: (1) provide new data that were hitherto not available to the scientific community, thereby facilitating further studies based on this proxy; (2) identify the dominant drivers of d13C over South America, allowing a first interpretation of the full dataset; (3) highlight the potential of this proxy for future paleoclimatic, paleoenvironmental and geochemical studies.

RC: I feel concerned by the fact that if temperature influence is eventually discarded, there is little evidence of what would be the expression of a local hydroclimate and the associated vegetation types. Indeed no information was given by the authors about the vegetation cover that is supposedly at the origin of the d13C and the absence of calibration makes the demonstration rather poor. When I started to look for such information through the original publications, I found that vegetation was defined at only 3 records upon 25 (I had no access to Utida et al 2020) mixing up biome with

composition of the vegetation type or physiognomy. For example the record of Tamboril is located within a Â'n Native semi-deciduous forests: : :ÂËŹz, Paraíso is Â'n densely covered by rainforest ÂËŹz and Mata Virgem is Â'n located in the Eastern regions of the tropical Savannah known as 'Cerrado' and the Amazon Forest ÂËŹz. For the last one, it is impossible to know whether the Cerrado or the Amazon forest is characterizing the area. These are two different biomes, and the reader would like to know about the vegetation type that grows above the cave, as for instance gallery forest, grassland with Cactaceae, campo limpo: : :etc).

Answer: We wish to emphasize that the main conclusion of our paper is that Prior Calcite Precipitation (PCP) is the main driver of d13C variability in stalagmites (note that this was also confirmed by reviewer 1). PCP is related to the local hydrology, but not to vegetation. Thus, we see no reason to provide such a detailed discussion regarding the vegetation at each individual site, since it plays at best a minor role our interpretation. During wet events the PCP rate decreases, which results in more negative d13C values in the stalagmites. Also, during wet periods decomposition of organic matter in the soil is enhanced, which provides more depleted values to the seepage solution, and therefore, to the stalagmites. Thus, the hydrological effects on vegetation will only reinforce the isotopic results in stalagmites derived by PCP, but this process is not related to specific species or types of plants.

RC: Under such poor descriptions of the vegetation cover, I have difficulties to believe the conclusion Â'n Most locations were dominated by C3 plants over the last two millennia ÂËŹz.

Answer: As discussed on lines 84-86 (from the original submission), geochemical models (Dreybrodt, 1988; Baker et al., 1997; McDermott, 2004) applied to stalagmites, using d13C values for the bedrock of +1‰ (typical values for South American carbonates) predict values below -6‰ for stalagmites where the predominant vegetation is composed of C3 plants. Considering that most of the stalagmites presented in this study have absolute values below -6‰ we believe that the argument that most locations were dominated by C3 plants is valid. This result is consistent with most d13C studies performed in Brazilian soils that also show a predominant composition of C3 plants over the country during the last 2 ka (Pessenda et al., 1996, 1998, 2004, 2005; Freitas et al., 2001; Schel-Ybert 2003; Calegari et al., 2013, 2017). However, we have improved this discussion in the manuscript to clarify this aspect.

RC: Finally the authors concluded that the dataset was able to show that _18O and _13C generally co-varied except at Tamboril cave and did not discuss the soil richness which was first expressed as a significant factor for @13C.

Answer: This is not our conclusion. We present the correlation between d18O and d13C for each stalagmite in Table A1. In some stalagmites the two isotopes are correlated, while in others they are not. Hence the Tamboril cave record is not an exception. The reasons why the two isotopes are correlated are discussed on lines 92-102 (from the original submission). They are mainly related to the fact that the amount effect influences the d18O values while PCP drives the d13C, and both these forcing are related to rainfall amount. The soil richness plays no role in this process.

RC: Neither was discussed the composition of the local/regional vegetation types that grow today above the cave.

Answer: We discuss this on lines 203-214 (from the original submission) and we present the vegetation map in Figure 1.

RC: Information are mixed up and some main points presented at the beginning of the ms are simply abandonned when reaching the discussion part. For instance, at Jaragua cave a decrease of 9 ‰ in the _13C values was interpreted as Â'n resulting from a combination of changes above the cave, including: changes in the predominant vegetation type from C4 to C3, increase of organic matter and soil horizons ÂËŹz which does not bring any strong information about the results.

Answer: In our introduction (section 1 and 2) we provide the reader with a complete

review about the mechanisms that drive the d13C variability in stalagmites, as well as a complete review of all studies that already used d13C in stalagmites from South America. We believe that this information is important since we are introducing a new stalagmite d13C dataset that was never interpreted before. However, some of the information provided from the previous papers (which focused on multiple time scales) does not applie to the new dataset with respect to the last 2ka. The example of Jaraguá cave was discussed in the introduction, among other examples, only to contextualize the d13C studies in stalagmites from South America.

RC: Moreover the breakdown between @18O and @13C after 1750 CE could also be due to deforestation and/or high fire activity.

Answer: This is indeed an interesting suggestion. We have reprhased this sentence, incorporating this suggestion. Now the sentence reads "...the hydrologic variability inferred from PC1-$\delta$13C is biased by vegetation changes, which respond to other influences beyond the local rainfall amount, such as temperature, atmospheric CO2, deforestation and fire, parameters that have increased significantly over the last 250 years."

RC: A comprehensive bibliography about the vegetation change and fire history of the different study areas is missing for the discussion. This paper needs a complete revision adding more precise information to support the interpretations and the conclusions.

Answer: We provide an exhaustive review regarding the mechanisms driving the d13C variability in stalagmites and a complete review covering all studies about d13C in stalagmites from South America. The vegetation changes are only one of many possible forcings behind the d13C variability in the stalagmites, and the main conclusion of our paper is that the d13C values are responding to changes in PCP rate, related to local hydrology (which is independent of vegetation changes). Thus, we see no reason to provide a detailed bibliography of vegetation (or fire) changes, since this is not the focus of our paper. Furthermore, the literature indicates that vegetation did not undergo significant changes during the last 2 ka in South America (Pessenda et al. 1996, 1998, 2004, 2005; Freitas et al., 2001; Schel-Ybert, 2003; Calegari et al., 2013, 2017). We wish to emphasize that our interpretations and conclusions are well supported by previous studies that we discuss in the introduction (section 1 and 2), including papers regarding d13C values in speleothems from South America as well as other tropical locations around the world.

RC: A proper discussion would also include the description and calibration of the vegetation and their influence on the obtained results. For these reasons I do not recommend this ms for publication.

Answer: The dataset presented by us in this paper is the result of tens of years of research made by different research group hosted at different institutions around the world. The presented study sites are from regions that are often very difficult to access, such as remote locations in the interior of the Amazon basin or high elevation sites in the Andes. To obtain calibration studies from all these sites is almost impossible and is far beyond the scope of this paper. Again, we wish to stress that vegetation changes play only a minor importance in our interpretations. We wish to highlight that the main aim of this paper is to provide and to describe a new dataset. We believe that this dataset contains valuable material for studies about geochemistry, paleoclimatology, paleoenvironment, caves, as well as for future calibration studies. By employing the data we make available here, other researchers can improve upon our study or provide alternative interpretations regarding the d13C values in stalagmites. We believe that this aspect alone is relevant for publishing this dataset and making it available to the scientific community.

RC: Specific issues and comments -Line 60 : importance of the tree roots : what about the herbs ?

Answer: We replaced the word (tree) by "plant".

RC: -Line 86 what about the CAM plants which are abundant in some regions covered by the data set ?

Answer: We added the sentence "... the presence of CAM plants can complicate this interpretation since this type of plant presents a large range of $\delta$13C values."

RC: -Line 127 last glacial

Answer: We have corrected this error.

RC: -Most of the sites located in the Caatinga show high abundance of Cactaceae which are C3 plants but do not represent dense forest neither rich organic soils. This should be mentionned and taken into account in the discussion.

Answer: We removed the sentence suggesting specific isotopic values for Caatinga and Cerrado. We believe this sentence was sowing confusion rather than aiding in the discussion, since Caatinga and Cerrado both feature C3 plants.

RC: -Comparisons with d18O records show that there is little difference with the d13C records regarding the climate interpretation and the authors come to the same conclusions except for the last centuries. Consequently, the gain of the d13C analyses in speleothem records is not clear.

Answer: d18O values in speleothems are generally considered to be proxies for climate interpretations. As we show in our manuscript, d18O and d13C have different drivers, yet the fact that both isotopes are correlated indicates that d13C can also be used as an important proxy for climate science. Our paper is the first to discuss both isotopes in a large set of data from South America, which makes our results novel and relevant to the speleothem community. Similar studies have been carried out in other regions of the world with differing results, mainly because d18O is not considered a proxy for the hydrological conditions outside the tropics, and because the strong temperature seasonality at high latitudes strongly affects the d13C in stalagmites (see Fohlmeister et al., 2020). In this way our results are unique.

RC: -Some interpretation/conclusion sound fuzzy. For instance at Jaragua cave a decrease of ‰ in the _13C values was first interpreted as Â′n resulting from a combination of changes above the cave, including changes in the predominant vegetation type from C4 to C3, increase of organic matter and soil horizons ÂËŹz. Later in the paper the vegetation hypothesis is abandonned and the decrease was finally related to temperature and atmospheric CO2.

Answer: We discussed this example in our introduction (section 2.2) as part of the overall review of prior studies related to d13C and speleothems in South America. This example was taken from Novello et al. (2019), but in the same paragraph (lines 123-142 from the original submission) we also present results from other studies that come to different conclusions. However, we agree that this discussion may have created some confusion. We have therefore removed part of this discussion to emphasize our main interpretation that is related to PCP.

RC: -How do we know that Â′n Most locations were dominated by C3 plants over the last two millennia and are characterized by speleothem _13C values more depleted than -6 ‰ ÂËŹz as no description of the vegetation is given? Also Line 87 Â′n Variations in soil _13C values and their evolution over time are controlled by carbon inputs from vegetation, which is proportional to the organic matter amount and vegetation density; ÂËŹz was promising but without any description of the vegetation it is difficult to evaluate the influence of the plant composition on the @13C.

Answer: See our answer above. This explanation is also included on lines 81-86 (from the original submission).

RC: -Line 209-213 Cerrado is also a tropical forest so what do you mean by Â′n Tropical Forest ÂËŹz ? also in the legend of figure 1 Rainforest/Atlantic forest, Atlantic forest is a biome that includes many vegetation types among them the rainforest (with no majuscule). What is the legend referring to ? Also line 289 what do the authors mean by Â′n a tropical forest ÂËŹz ? is it a rainforest or a seasonal forest or a dry forest or a

semi-deciduous forest or a cloud forest?

Answer: Cerrado is defined as a Brazilian Savanna and/or a tropical Savanna (Ruggiero et al., 2002; Jepson, 2005; Brannstrom et al., 2008). Tropical forests in Brazil are characterized by Amazon and Atlantic forests as is indicated in Figure 1 and its legend, defined by Olson et al. (2001). This is the standard nomenclature used in many papers (Keller et al., 2001; Rocha et al., 2009; Alvez et al., 2010; Xaud et al., 2013).

RC: -Line 211 the 3 sites that are called Â′n andean sites ÂËŹz are located in very different environmental conditions : Huagapo, 3850 m asl should correspond to a high elevation grassland as the tree line never climbed further up than _3500 m in the last 3000 years, Umajalanta 2650 m asl Â′n monsoon-related convection and condensation over the Amazon Basin ÂËŹz which corresponds to a cloud forest, Palestina, 870 m asl, the transition between rainforest to cloud forest.

Answer: We do not discuss Andean sites on line 211 because it is not our intention to describe all sites individually. In this paragraph, we only present the dominant climate/vegetational settings over South America.

RC: -Line 228-230 Are you saying that Cerrados and rainforest have the same d13C ? Â′n The vegetation domains of Tropical Forests (Rainforest/Atlantic forest) and Cerrado include the speleothems with the lowest _13C values (mean of -8.9 ‰ and -8.5 ‰ respectively),.. ÂËŹz

Answer: No, this is not what we meant to express with this sentence. We have removed this sentence from the manuscript to avoid misunderstanding.

RC: -Line 232 Appendix Fig A2 What is meant by highland records ? they are located in three different vegetation covers (see above). Then the @13C should be different for each of these vegetation types. Why isn't it so ?

Answer: We have removed this figure from our paper. We realize that this figure did show confusion, rather than adding new information to our discussion.

RC: -Line 235-238 then it is not correlated with vegetation as it was explained at the beginning of the manuscript? Answer: Yes. It is not correlated with vegetation. At the beginning of the manuscript we presented a literature review, which is not necessarily consistent with our own analysis and interpretation of the data.

RC: -Line 248 etc MCP-PCA could also include the degree of opening of the vegetation types that grow above the cave in the discussion.

Answer: In this paragraph, we only describe the results. The interpretation of the data is in the section 'Discussion; on lines 304-323 (from the original submission). There we mention this aspect.

RC: -Line 254 this could have been shown also with d18O ?

Answer: Yes, we show this in both isotopes in Figure 4 and discuss this aspect on lines 314-323 and lines 331-337 (from the original submission).

RC: -Line 278 denser vegetation : this could be checked

Answer: We replaced this statement with "increase of vegetation density".

RC: -Line 290 I disagree here because in the Caatinga there is no C4 plants, a high density of C3 deciduous trees plus the Cactaceae (C3 carbon metabolism).

Answer: We did not mention in this paragraph that the Caatinga includes C4 plants. To avoid confusion, we have removed this sentence, as it was not considered relevant for our discussion.

RC: -Line 291 TR5 add the vegetation type/ the environment of the cave

Answer: We did not provide the vegetation type in this instance, because we do not want to mislead the reader in believing that the d13C values from TR5 are related to vegetation. As we document in this paragraph, they are more related to hydrological conditions.

RC: -Line 291-303 : the d13C relates to climate and evaporation ? not anymore with soil thickness ? and what about plant assemblages ? these last two points are not discussed (also line 333) when they were listed as main factors of d13C variability earlier in the ms.

Answer: As can be seen from our Discussion, the variability in d13C values in our dataset is mainly related with hydroclimate, and not vegetation or soil thickness. Furthermore, significant changes in vegetation and soil thickness are not expected to have occurred during the last 2 ka, as these processes require more time to take place.

RC: -Line 311 C3 plants do show a broad range of physiognomies in the Tropics Line 311: this was not really demonstrated in the ms neither in Novello et al 2019

Answer: The plant physiognomies are not relevant for our analysis. Our study is focused on geochemical proxies in stalagmites. Vegetation processes play only a minor role in our interpretation, as mentioned above.

RC: -Line 313 If d13C reflects hydroclimate, why analysing the Carbon stable isotope as @d8O proved to be efficient in that field?

Answer: We are exploring a new proxy with several implications for climate, environment, cave and geochemical studies. As we state above: "Our paper is the first to discuss both isotopes in a large set of data from South America, which makes our results novel and relevant to the speleothem community. Similar studies have been carried out in other regions of the world with differing results, mainly because d18O is not considered a proxy for the hydrological conditions outside the tropics, and because the strong temperature seasonality at high latitudes strongly affects the d13C in stalagmites (see Fohlmeister et al., 2020)". In this way our results are unique.

RC: -Line 334 after 1750 CE, may be consider deforestation ? which could eventually introduce a discussion on the vegetation cover ?

Answer: Please see our answer above. We have changed our statement regarding this

aspect.

RC: Line 336 which vegetation changes and where ?

Answer: It is clear that we are discussing PC1 in this instance. We simply list possible influences that may affect the entire dataset.

RC: Line 337 vegetation responds to other influence: : :yes I thought it was soil thickness ? Although here it is the temperature that is inferred.

Answer: We list possible influences on vegetation in a continental scale, which could affect all sites simultaneously. Soil thickness is a local influence.

RC: Line 349 so again not related to vegetation

Answer: See our answers above. Our paper is not about vegetation. The vegetation is only one of many aspects possibly influencing our proxy.

RC: Line 354 This could be an important issue to consider for instance when comparing with other compilation Line 354 356 I do not agree with this conclusion. The predominance of C3 plants on the study sites was not shown.

Answer: See our comments above about this statement.

RC: Line 363 365 what is the novelty here ? it was already showed by d18O?

Answer: See our answers above. We are exploring a new proxy and providing a new dataset to the scientific community composed of data collected over the entire South American continent. This is why we submitted this dataset for publication to ESSD; a journal focused on publishing novel data sets.

RC: Line 366-367 and deforestation ?

Answer: See our answer above. We provide a comment about deforestation in the discussion.

RC: Tables and Figures Table 1 Add at least the biome in front of each site and the

vegetation type that grows above the cave.

Answer: This information is already contained in Figure 1. We do not have space to include an additional column in this table.

RC: Table A1 add the length of the speleothem

Answer: This information is not relevant for the data interpretation and is not discussed anywhere in our manuscript.

RC: Figure 1 Legend figure 1 : Cerrados with S

Answer: The correct word is Cerrado (without S), as provided in the text.

RC: Figure 2 what don't we see 25 points ?

Answer: Please see our answer provided to reviewer 1. Some of the stalagmites are from the same cave and/or are from regions with the same climatic conditions. These stalagmites were merged and are represented by a single point. We have improved the description of this correlation in the Methods section..

RC: Figure A1 is not lisible

Answer: This figure aims to illustrate the range of the d13C values covered by South America's speleothems. For this purpose, the data organization as presented, is the best way to show all records together.

RC: Figure A2 : what do you call a vegetation domain ? a biome ? a vegetation type ? anyway this is none of the category represented here. It is not possible to relate the d13C to the vegetation cover. This point needs further development in your argumentation.

Answer: We grouped the stalagmites growing under similar or closely related vegetation conditions to see if there are isotopic differences between these groups. However, this figure did not reveal any significant differences. Thus, we decided to remove it from

the manuscript.

References:

Alves, L.F., Vieira, S.A., Scaranello, M.A., Camargo, P.B., Santos, F.A.M., Joly, C.A., Martinelli, L.A.: Forest structure and live aboveground biomass variation along an elevational gradient of tropical Atlantic moist forest (Brazil). Forest Ecol. Manag., 260, 679-691, 2010.

Baker, A., Ito, E., Smart, P.L. and McEwan, R.F.: Elevated and variable values of d13C in speleothems in a British cave system. Chem. Geol., 136, 263–270, 1997.

Brannstrom, C., Jepson, W., Filippi, A. M., Redo, D., Xu, Z., Ganesh, S.: Land change in the Brazilian Savanna (Cerrado), 1986-2002: Comparative analysis and implications for land-use policy. Land Use Policy, 25, 579-595, 2008.

Calegari, M.R., Madella, M., Vital-Torrado, P., Pessenda, L.C.R., Marques, F.A.: Combining phytoliths and soil organic matter in Holocene palaeoenvironmental studies of tropical soils: the example of an oxisol in Brazil. Quarter. Int., 287: 47-55, 2003.

Calegari, M.R., Madella, M., Brustolin, L.T., Pessenda, L.C.R., Buso Jr. A.A., Francisquini, M.I., Benassolli, J.A., Vidal-Torrado, P.: Potential of soil phytoliths, organic matter and carbono isotopes for small-scale differentiation of tropical rainforest vegetation: A pilot study from the campos nativos of the Atlantic Forest in Espírito Santo State (Brazil). Quarter. Int, 437: 156-164, 2017.

Dreybrodt, W.: Processes in Karst Systems. Springer Series in Physical Environment. Springer, Heidelberg, 282pp, 1988.

Fohlmeister, J., Voarintsoa, N. R. G., Lechleitner, F. A., Boyd, M., Brandstätter, S., Jacobson, M. J. and Oster, J.: Main controls on the Stable Carbon Isotope Composition of Speleothems. Geochim. Cosmochim. Acta, GCA11710, 2020.

Freitas, H.A., Pessenda, L.C.R., Aravena, R., Gouveia, S.E.M., Ribeiro A.S., Boulet,

R. Late Quaternary vegetation dynamics in the southern Amazon Basin inferred from carbon isotopes in soil organic matter. Quarter. Res, 55, 39-46, 2001.

Jepson, W.: A disappering biome? Reconsidering land-cover change in Brazilian savanna. The Geographical Journal, 171 (2), 99-111, 2005.

Keller, M., Palace, M., Hurt, G.: Biomass estimation in the Tapajos national Forest, Brazil Examination of sampling and allometric uncertainties. Forest Ecology and Management, 154, 371-382, 2001.

McDermott, F.: Palaeo-climate reconstruction from stable isotope variations in speleothems: a review. Quat. Sci. Rev., 23, 901–918, doi .org /10.1016 /j.quascirev.2003.06.021, 2004.

Novello, V. F., Cruz, F. W., McGlue, M. M., Wong, C. I., Ward, B. M., Vuille, M., Santos, R. A., Jaqueto, P., Pessenda, L. C. R., Atorre, T., Ribeiro, L. M. A. L., Karmann, I., Barreto, E. S.; Cheng, H., Edwards, R. L., Paula, M. S. and Scholz, D.: Vegetation and environmental changes in tropical South America from the last glacial to the Holocene documented by multiple cave sediment proxies. Earth Planet. Sci. Lett., 524, 115717, doi.org/10.1016/j.epsl.2019.115717, 2019.

Pessenda, L.C.R., Aravena, R., Melfi, A.J., Telles, E.C.C., Boulet, R., Valencia, E.P.E., Tomazello, M. 1996. The use of carbon isotopes (C-12, C-13) in soil to evaluate vegetation changes during the Holocene in Central Brazil. Radiocarbon, 38 (2), 191-201, 1996.

Pessenda, L.C.R., Gomes, B.M., Aravena, R., Ribeiro, A.S., Boulet, R., Gouveia, S.E.M. The carbon isotope record in soils along a forest-cerrado ecosystem transect: implications for vegetation changes in the Rondônia State, southwestern Brazilian Amazon region. The Holocene, 8: 599–603, 1998.

Pessenda, L.C.R., Ribeiro, A.S., Gouveia, S.E., Aravena, R., Boulet, R., Bendassoli, J.A.: Vegetation dynamics during the late Pleistocene in the Barreirinhas region,

Maranhão State, Northeastern Brazil, based on carbon isotopes in soil organic matter. Quarter. Res., 62, 183–93, 2004.

Pessenda, L.C.R., Ledru, M.P., Gouveia, S.E.M., Aravena, R., Ribeiro, A.S., Bendassolli, J.A., Boulet, R.: Holocene palaeoenvironmental reconstruction in northeastern Brazil inferred from pollen, charcoal and carbon isotope records. The Holocene, 15: 814-22, 2005.

Rocha, H.R., Mazi, A.O., Cabral, O.M., Miller, S.D., Goulden, M.L., Saleska, S.R., R.-Coupe, N., Wolsy, S.C., Borna, L.S., Artaxi, P., Vourlitis, G., Noguiera, J.S., Cardoso, F.L., Nobre, A.D., Kruijt, B., Freitas, H.C., Von Randow, C., Aguiar, R.G., Maina, J.F.: Patterns of water and heat flux across a biome gradient from tropical forest to savanna in Brazil. J. Geophys. Res., 114, G00B12, 2009.

Ruggiero, P.G.C.; Batalha, M.A.; Pivello, V.R., Meirelles, S.T.: Soil-vegetation relationship in cerrado (Brazilian savanna) and semideciduous forest, Southern Brazil. Plant. Ecology, 160, 1-16, 2002.

Scheel-Ybert, R., Gouveia, S.E.M., Pessenda, L. C. R., Aravena, R., Coutinho, L.M., Boulet, R.: Holocene palaeoenvironmental Evolution in the São Paulo State (Brazil), based on anthracology and soil d13C analysis. The Holocene, 13 (1), 73-81, 2003.

Xaud, H.A.M., Martins, F.S.R., Santos, J.R.: Tropical forest degradation by mega-fires in the northern Brazilian Amazon. Forest Ecol. Manag., 294, 97-106, 2013.